# ToF-IP: Time-of-Flight Enhanced Sparse Inertial Poser for Real-time Human Motion Capture

**Yuan Yao[1]**    **Shifan Jiang[1]**    **Yangqing Hou[1]**    **Chengxu Zuo[1]**    **Xinrui Chen[1]**

**Shihui Guo[1]\***                                **Yipeng Qin[2]**

[1] School of Informatics, Xiamen University, China
[2] School of Computer Science & Informatics, Cardiff University, UK

## Abstract

Sparse inertial measurement units (IMUs) provide a portable, low-cost solution for human motion tracking but struggle with error accumulation from drift and sensor noise when estimating joint position through time-based linear acceleration integration (i.e., *indirect* measurement). To address this, we propose ToF-IP, a novel 3D full-body pose estimation system that integrates Time-of-Flight (ToF) sensors with sparse IMUs. The distinct advantage of our approach is that ToF sensors provide *direct* distance measurements, effectively mitigating error accumulation without relying on *indirect* time-based integration. From a hardware perspective, we maintain the portability of existing solutions by attaching ToF sensors to selected IMUs with a negligible volume increase of just 3%. On the software side, we introduce two novel techniques to enhance multi-sensor integration: (i) a Node-Centric Data Integration strategy that leverages a Transformer encoder to explicitly model both intra-node and inter-node data integration by treating each sensing node as a token; and (ii) a Dynamic Spatial Positional Encoding scheme that encodes the continuously changing spatial positions of wearable nodes as motion-conditioned functions, enabling the model to better capture human body dynamics in the embedding space. Additionally, we contribute a 208-minute human motion dataset from 10 participants, including synchronized IMU-ToF measurements and ground-truth from optical tracking. Extensive experiments demonstrate that our method outperforms state-of-the-art approaches such as PNP, achieving superior accuracy in tracking complex and slow motions like Tai Chi, which remains challenging for inertial-only methods.

## 1 Introduction

Sparse inertial measurement units (IMUs) have emerged as a promising solution for human motion tracking due to their portability, low cost, and camera-free nature [8, 11, 20, 42]. However, despite their potential, sparse IMUs face inherent numerical challenges due to their *indirect* method of position estimation. Specifically, sparse IMUs estimate velocity and position by time-based linear acceleration integration, a process highly prone to error accumulation from drift and sensor noise. These errors are further amplified by the task's reliance on human body forward kinematics, where positional inaccuracies in intermediate joints propagate along the kinematic chain, leading to greater errors at terminal joints. Consequently, accurately tracking subtle positional changes of key joints during low-velocity motions (where the motion-signal-to-noise ratio is low) and enhancing the long-term stability of sparse IMU systems remain persistent and critical challenges in this field [37].

---

*Corresponding author.

39th Conference on Neural Information Processing Systems (NeurIPS 2025).

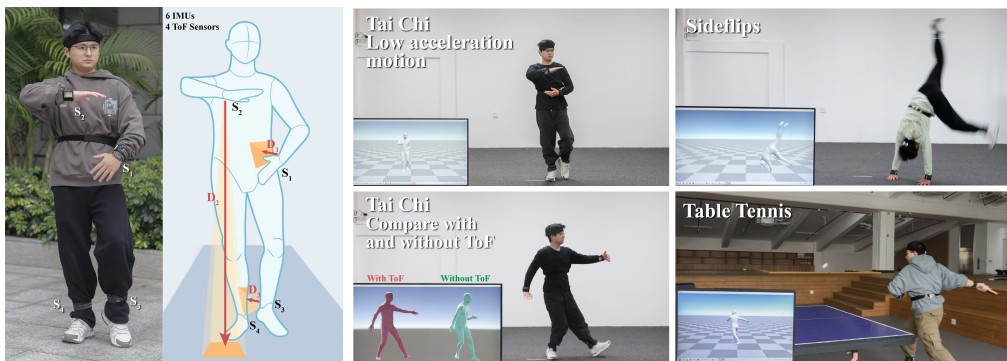

Figure 1: ToF-IP integrates distance maps from four Time-of-Flight (ToF) sensors to overcome the drift and error accumulation inherent in inertial-only motion capture, enabling more accurate and stable motion tracking even for challenging motions such as slow, controlled sequences (e.g., Tai Chi) and rapid, complex actions (e.g., sideflips).

To date, most existing methods focus on providing software-based solutions to the aforementioned challenge. For example, deep learning approaches, such as bidirectional RNNs, have been developed to regress IMU data into pose sequences [8]. Subsequent RNN-based methods further improved pose prediction accuracy and integrated global position estimation [36], with some incorporating more precise dynamic models [37]. Beyond RNNs, alternative architectures have emerged, including attention-based models for capturing physical motion during stationary phases [11]. Nevertheless, software solutions are inherently limited, as they still rely on *indirect* pose estimation and can only mitigate, rather than fully resolve, the problem of error accumulation.

In this paper, we present ToF-IP, a novel 3D full-body pose estimation system that addresses the abovementioned challenge by integrating Time-of-Flight (ToF) distance sensors with sparse IMUs. The distinct advantage of our approach is that ToF sensors provide *direct* distance measurements without relying on *indirect* time-based integration, offering learnable solution space constraints for the position of limb-end joints. A key innovation of our ToF-IP lies in its hardware design, which maintains the lightweight and portable nature of existing sparse IMU systems. Specifically, without altering the standard 6-IMU layout, we incorporate 4 highly integrated ToF sensors based on single-photon avalanche diodes (SPAD) [26] directly onto the IMU circuit boards equipped at left forearm, right forearm, left lower leg, and right lower leg of the human body, ensuring minimal impact on wearability. On the software side, we fully harness the potential of ToF-inertial sensing by proposing a unified Transformer-based framework with two key innovations. First, we introduce a Node-Centric Data Integration strategy that explicitly captures the hierarchical structure of multi-sensor data. Unlike prior methods that flatten all sensor inputs into a single vector-wise organization (discarding the spatial and structural semantics of node), our approach represents each sensing node as an independent token. This token is constructed through intra-node data integration of ToF depth, IMU acceleration and orientation data. These node tokens are then contextually integrated using the self-attention mechanism of a Transformer encoder, which naturally facilitates inter-node communication and dynamic weighting based on task-relevant dependencies. Second, we propose a Dynamic Spatial Positional Encoding (Dyn-PE) method tailored to the unique challenges of wearable sensing. Unlike traditional positional encodings in NLP or vision tasks that assume static or grid-based positions, the spatial configuration of wearable nodes evolves continuously with human motion. To capture this, Dyn-PE models each node's position as a learnable function of global motion signals, generating time-varying encodings that reflect the node's physical displacement in space. This dynamic encoding enhances the model's spatial awareness, allowing it to better resolve ambiguous interactions and motion patterns across nodes. Extensive experimental results show that, compared to state-of-the-art (SOTA) methods, our approach significantly improves joint position estimation, achieving superior accuracy in tracking complex and slow movements like Tai Chi.

In summary, our contributions include:

- We design an in-situ enhanced bimodal wearable sensing platform for 3D full-body tracking, retaining the conventional layout of 6 sensing nodes. The platform allows for the flexible use of either single IMU sensing or IMU+ToF bimodal sensing, with only a 3% increase in volume.

- We propose *ToF-IP*, a novel Transformer-based inertial-ToF motion capture framework that introduces two key innovations on the software side: (i) a Node-Centric Data Integration strategy that preserves the structural semantics of multi-sensor data by treating each sensing node as a token and hierarchically integrating intra- and inter-node information via self-attention; and (ii) a Dynamic Spatial Positional Encoding scheme that models the continuously evolving spatial positions of wearable nodes as motion-conditioned functions, enhancing spatial awareness and robustness to body movement variations. Our approach demonstrates substantial improvements over state-of-the-art methods, delivering higher positional precision and more accurate joint angle estimation, particularly in the upper limbs and legs.

- We propose *ToF-IP-DB*, a large dataset containing over 20 types of motion activities, 208 minutes (749,000 frames) collected from 10 participants (3 male, 7 female), including dynamic motions such as dances and aerobics, as well as slow-paced movements like Tai Chi and Baduanjin. This dataset uniquely combines synchronized ToF distance maps, 6-DoF IMU signals, and SMPL reference poses, with GT motion data.

## 2 Related Work

### 2.1 Pose Estimation Using Inertial Sensors

With the rapid advancement of MEMS technology [9], IMUs (Inertial Measurement Units) have become smaller, more power-efficient, and affordable. This has led to numerous works leveraging IMUs for human pose estimation. Despite their independence from external environments, the working principle of IMUs—using accelerometers, gyroscopes, and magnetometers to compute orientation—limits their accuracy. In the commercial market, motion capture systems employing 17–19 IMUs for human pose estimation exist [35, 23], but they require operation in uniform magnetic field environments and have limited usage durations, as accumulated integration errors can lead to model collapse.

The pursuit of lightweight solutions has spurred research into using sparse IMUs—typically six sensors placed at limb extremities. The advent of the SMPL [18] and AMASS [19] datasets has enabled the creation of large-scale mocap/IMU-aligned datasets. Synthetic continuous acceleration and rotation data were generated by placing virtual IMUs on specific body parts in the AMASS dataset, facilitating pose estimation in both offline [31] and real-time settings. Deep learning methods, such as bidirectional RNNs, were designed to regress IMU data to pose sequences [8]. Building on this, RNN-based approaches have improved pose prediction accuracy and incorporated global position estimation [36], with some integrating more precise dynamic models, such as those in [37]. Other methods explore alternative network architectures, including attention-based models for learning physical motion during stationary points [11] and spatiotemporal modules for more accurate pose estimation [34]. Efforts have also been made to enhance comfort and convenience. For example, some methods use VR headsets, smartphones, and other wearable devices to estimate upper-body poses and predict lower-body movements. Zuo et al. integrated IMUs into loose clothing [42], effectively regressing upper-body poses while mitigating artifacts caused by fabric-induced jitter.

Despite such success, challenges remain in pose estimation using IMUs alone [22, 10, 14, 28]. These include IMUs' inability to directly measure velocity or position, accumulation of drift errors [15, 13], and reliance on forward kinematics models [25] composed of bones and joints. Sparse IMU layouts—especially those at the extremities of limbs, such as hands and legs—face difficulties due to the higher degrees of freedom and heavier prediction tasks borne by individual sensors.

### 2.2 Time-of-Flight Distance Sensors

To mitigate these issues, recent research has explored hybrid solutions that integrate additional sensor modalities [1, 3, 17]. One common strategy is to incorporate global positioning or localization techniques to provide absolute positional constraints [7]. For instance, Zihajehzadeh et al. [41] combined IMUs with ultrawideband (UWB) localization to eliminate yaw angle drift in lower-body tracking, leveraging UWB's absolute position measurements to correct inertial estimates. Similarly, Liu et al. [16] used a micro-flow sensor to estimate motion velocity, enabling accurate extraction of gravitational acceleration from accelerometer data and improving posture tracking stability.

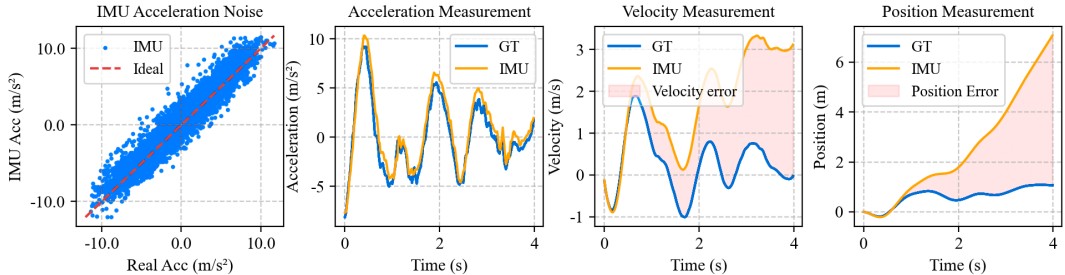

Figure 2: Illustration of error accumulation in velocity and position. Even slight noise in IMU acceleration leads to significant integration errors in velocity and position within just 4 seconds. Acceleration signals are taken from the TIC dataset [43], with ground-truth provided for comparison.

Another class of methods exploits vision-based and depth sensors to compensate for inertial drift [30, 32, 5, 21, 38]. Depth sensors such as LiDAR and structured-light cameras have been explored for markerless tracking. ToF sensors broadly refer to methods for precise distance measurement based on the time taken by light pulses or continuous waves to travel [6, 12]. Unlike vision-based methods, ToF sensors are resistant to variations in lighting and occlusions, providing robust depth measurements even in challenging conditions [33, 24]. Some studies [27, 4] have integrated ToF sensors into external environments for global pose estimation, but their application in wearable systems remains underexplored.

By carefully balancing factors like power consumption and heat dissipation, we augment the conventional six-IMU sparse layout with four low-resolution depth Time-of-Flight (ToF) sensors. This hardware integration enhances inertial pose estimation by capturing inter-limb distances and contact points with the environment, providing real-time, in-situ constraints that mitigate positional drift while preserving the wearability and portability characteristic of traditional inertial tracking systems.

## 3   Inherent Limitation of Sparse Inertial-only Motion Capture

Under a sparse IMU configuration, only a subset of joint orientations can be directly measured. To compensate for missing measurements, acceleration data is commonly used as an additional input [8], as it carries implicit cues about joint positions that can aid in inferring unobserved joint orientations. In principle, joint positions can be obtained by double-integrating the acceleration signals over time. However, in practice, real-world acceleration measurements are prone to various sources of error, such as sensor noise and signal drift, inevitably resulting in significant error accumulation over time:

**Proposition 3.1** (Error Accumulation Analysis). *Following standard statistical practice, we assume that the acceleration measurement error at any timestamp $\tau \in (0, t)$ follows a normal distribution $\epsilon_a(\tau) \sim \mathcal{N}(\mu, \sigma^2)$. Then, we have:*

- *Distribution of joint velocity error: $\epsilon_v(t) \sim \mathcal{N}(\mu t, \sigma^2 t)$*

- *Distribution of joint position error: $\epsilon_s(t) \sim \mathcal{N}(\frac{1}{2}\mu t^2, \frac{1}{2}\sigma^2 t^2)$*

*Proof.* The proof is provided in Section 2 of the supplementary material. □

Proposition 3.1 and Fig. 2 shows that the joint position error grows quadratically over time, highlighting an inherent limitation of sparse inertial-only solutions.

## 4   Method

To address the inherent limitation of sparse inertial-only solutions discussed in Sec. 3, we propose integrating Time-of-Flight (ToF) sensors, which provide *direct* distance measurements to mitigate error accumulation from time-based integration.

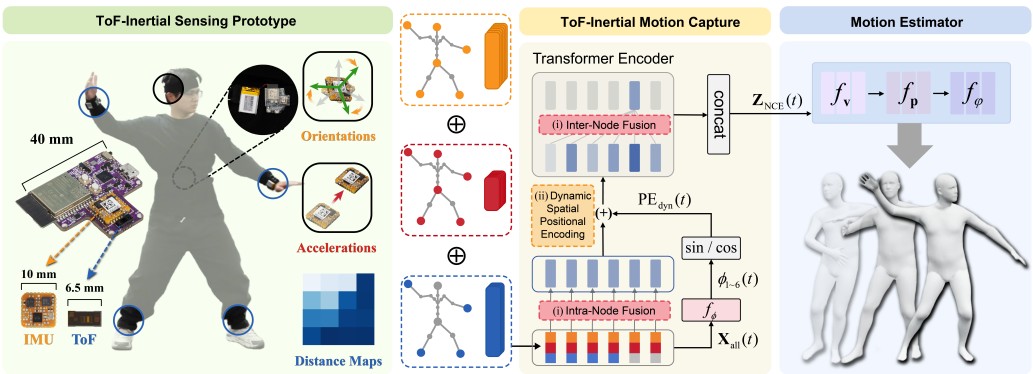

Figure 3: Illustration of Our ToF-Inertial Sensing Prototype and Motion Capture Method. **Left:** Our ToF-Inertial sensing prototype comprises 6 IMU-attached sensing nodes, with 4 of them (located on the left / right forearm and left / right lower-leg) additionally integrated with ToF sensors. **Middle:** Our ToF-Inertial motion capture method comprises two novel techniques: (i) a node-centric data integration strategy based on a Transformer encoder, encompassing intra-node and inter-node data integration between ToF-Inertial sensor; (ii) a dynamic spatial positional encoding to adapt to the dynamic changes in the spatial positions of sensing nodes during motion capture; **Right:** Three sequentially connected LSTM networks ($f_\mathrm{v}$, $f_\mathrm{p}$, $f_\phi$) serve as motion estimators to transform the integrated sensing node data into human body movements.

## 4.1 ToF-Inertial Sensing Prototype

As shown in Fig. 3 (left), to maximize compatibility with existing inertial-only solutions and minimize impact on wearability, i) we adopt the standard 6-node layout used in prior works [36, 37, 11, 39] and place 6 IMUs on the left/right forearms, left/right lower legs, pelvis and head, respectively; ii) we integrate 4 ToF sensors into the IMU nodes on the left/right forearms and left/right lower legs. Specifically, we mount the ToF sensors on the inner wrists and rear ankles to capture distance measurements from distal joint endpoints to nearby body parts and the ground. Note that we omit ToF sensors from the head and pelvis nodes, as these positions seldom observe relevant surfaces.

## 4.2 ToF-Inertial Motion Capture

**Overview.** As shown in Fig. 3 (middle, right), our ToF-Inertial Motion Capture framework adopts a Transformer-based architecture that incorporates two key innovations for integrating IMU and ToF data: (i) Node-Centric Data Integration, which explicitly models intra- and inter-node interactions through tokenized node representations; and (ii) Dynamic Spatial Positional Encoding, which encodes the time-varying spatial positions of sensing nodes using motion-conditioned functions. Following [36, 37, 39], we further employ a cascade of three LSTM networks as motion estimators.

### 4.2.1 Node-Centric Data Integration

**Conditions for Effective ToF Integration.** Although the direct distance measurements provided by ToF establish a data foundation for improving joint position estimation, their effectiveness hinges on two key data integration conditions:

- **[Intra-node Integration]** Each ToF sensor must integrate orientation and acceleration data from its co-located inertial sensor within the sensing node to determine the viewing direction and motion state information;
- **[Inter-node Integration]** Since both the motion of the sensing node itself and the captured object can cause changes in the ToF distance map, integrating measurements from other sensing nodes is required to distinguish absolute motion from relative motion.

**Limitation of Existing Data Integration Method.** Existing methods typically use fully-connected networks (FCNs) for data integration. This approach directly flattens data from multiple nodes into a single vector, losing the inherent data structure organized by sensing nodes and overlooking

intra-node integration. Formally, consider an embedded feature $f_i$ from an FCN layer, we have:

$$f_i = \text{Flatten}(\mathbf{x}^{(1)}, \ldots, \mathbf{x}^{(N)}) \cdot W_i + b_i \tag{1}$$

where $\{\mathbf{x}^{(1)}, \ldots, \mathbf{x}^{(N)}\}$ are the data of the $N$ nodes. In this formulation, the linear combination of $\mathbf{x}^{(j)}, j = 1, \ldots, N$ inherently achieves inter-node integration, but overlooks the intra-node data integration within each $\mathbf{x}^{(j)}$.

**Node-Centric Data Integration.** To address this challenge, we propose replacing fully-connected networks used in previous methods with a Transformer Encoder. In this scheme, each sensing node is converted into an independent token, with all encoding performed at the token (node) level. Consider the encoding process for an arbitrary output token $\mathbf{z}_T^{(n)}$ of node $n$, which unfolds in two structured steps to explicitly perform intra-node and inter-node data integration:

- *1) Intra-node Data Integration via Tokenization.* The multi-modal data of each node $\mathbf{x}^{(n)} = [d^{(n)}, a^{(n)}, R^{(n)}]$ is encoded into an intra-node token $\mathbf{z}_{\text{intra}}^{(n)} \in \mathbb{R}^{d_{\text{model}}}$ as follow:

$$\mathbf{z}_{\text{intra}}^{(n)} = f_T(\mathbf{x}^{(n)}) + \text{PE}^{(n)} \tag{2}$$

where $f_T$ is a tokenize function, $d_{\text{model}}$ is size of token. $d \in \mathbb{R}^{16}$, $a \in \mathbb{R}^3$, and $R \in \mathbb{R}^{3 \times 3}$ are ToF depth maps, IMU acceleration and orientation, respectively, $\text{PE}^{(n)}$ is positional encoding of the $n$-th sensing node. For IMU-only sensing node (head and hip), $d$ is set to zeros.

- *2) Inter-node Data Integration via Self-Attention.* The intra-node tokens $\{\mathbf{z}_{\text{intra}}^{(1)}, \ldots, \mathbf{z}_{\text{intra}}^{(N)}\}$ are then integrated via Transformer Encoder's self-attention mechanism, which computes inter-node interaction weights $A \in \mathbb{R}^{N \times N}$ applied to intra-node tokens:

$$\mathbf{z}_{\text{inter}}^{(i)} = \mathbf{z}_{\text{intra}}^{(i)} + \sum_{j=1}^{N} A_{ij} (\mathbf{z}_{\text{intra}}^{(j)} \cdot \mathbf{W}_{\text{v}}) \tag{3}$$

where $\mathbf{W}_{\text{v}}$ projects tokens into value space, and the $\mathbf{z}_{\text{intra}}^{(i)}$ is residual connection term. Then each $\mathbf{z}_{\text{inter}}^{(i)}$ will go through layer norm (LN) and feed-forward network (FFN) in the Transformer Encoder and concatenated to produce the final embedding vector $\mathbf{Z}_{\text{NCI}} \in \mathbb{R}^{(N \times d_{\text{model}})}$.

### 4.2.2 Dynamic Spatial Positional Encoding

**Static Positional Encoding.** As Eq. 2 shows, positional encoding PE is a fundamental process to incorporating position information into tokens. Existing static positional encoding methods assign **static** positional values $\phi$ to each token via an addition operation. For example, the static positional encoding in the original Transformer [29] is as follows:

$$\text{PE}_{\text{sta}}^{(n, 2i, 2i+1)} = [\sin(\omega_i \cdot \phi_{pos}), \cos(\omega_i \cdot \phi_{pos})], \quad \phi_{pos} = n, \quad \omega_i = 10000^{-2i/d_{\text{model}}} \tag{4}$$

where $n$ is the index of the input token, $d_{\text{model}}$ is the dimension of the token, $\phi_{pos}$ is the static positional value determined by the token index (sequential position). The $n$-th sensing node token processed by static positional encoding can then be represented as $\mathbf{z}_{\text{intra}}^{(n)} = f_T(\mathbf{x}^{(n)}) + \text{PE}_{\text{sta}}^{(n)}$, where $\text{PE}_{\text{sta}}^{(n)}$ is static positional encoding calculated by Eq. 4.

**Our Dynamic Spatial Positional Encoding.** However, unlike the static and discrete positions in natural language processing and computer vision applications (such as word order or pixel grid coordinates), the positions of sensing nodes are **dynamic** and continuous **spatial** positions that change with human body movements. Considering these characteristic, we model the positions of sensing nodes as continuous functions of motion signals and propose dynamic spatial positional encoding (Dyn-PE) as follows:

$$\text{PE}_{\text{dyn}}^{(n, 2i, 2i+1)}(t) = [\sin(\omega_i \cdot \phi_n(t)), \cos(\omega_i \cdot \phi_n(t))], \quad \phi_n(t) = f_\phi^n(\mathbf{X}_{\text{all}}(t)), \quad \omega_i = 10000^{-2i/d_{\text{model}}} \tag{5}$$

where $\phi_n(t)$ are the dynamic positional value of sensing node $n$ at time $t$ ($n = 1, 2, \ldots, 6$), $\mathbf{X}_{\text{all}}(t) = d^{1 \to 6}(t), a^{1 \to 6}(t), R^{1 \to 6}(t)$ are data of all six sensing nodes, serving as human motion signal. The $f_\phi^1, \ldots, f_\phi^n$ are position estimation functions implemented with a 2-layer MLP. Then, our Dynamic Spatial Positional Encoding can be integrated into Eq. 2 as: $\mathbf{z}_{\text{intra}}^{(n)} = f_T(\mathbf{x}^{(n)}) + \text{PE}_{\text{dyn}}^{(n)}(t)$.

### 4.2.3 Motion Estimators

Following [36, 37, 39], we feed the sensor data embedding $\mathbf{z}_{\text{NCI}}$ as the shared input to three sequential motion estimators for joint velocity $\mathbf{v} \in \mathbb{R}^{J \times 3}$, joint position $\mathbf{p} \in \mathbb{R}^{J \times 3}$ and joint rotation $\boldsymbol{\varphi} \in \mathbb{R}^{J \times 6}$:

$$\mathbf{v}(t) = f_{\mathbf{v}}(\mathbf{Z}_{\text{NCI}}(t)) \quad \mathbf{p}(t) = f_{\mathbf{p}}(\mathbf{Z}_{\text{NCI}}(t), \mathbf{v}(t)) \quad \boldsymbol{\varphi}(t) = f_{\boldsymbol{\varphi}}(\mathbf{Z}_{\text{NCI}}(t), \mathbf{p}(t)) \quad (6)$$

where $J = 18$ is the total number of tracked joints, $f_{\mathbf{v}}, f_{\mathbf{p}}, f_{\boldsymbol{\varphi}}$ are RNN-based motion estimators. These motion estimators are trained in a supervised manner using the following motion loss:

$$\mathcal{L}_{\text{motion}} = ||\mathbf{v}(t) - \mathbf{v}^{\text{GT}}(t)||_2^2 + ||\mathbf{p}(t) - \mathbf{p}^{\text{GT}}(t)||_2^2 + ||\boldsymbol{\varphi}(t) - \boldsymbol{\varphi}^{\text{GT}}(t)||_2^2 \quad (7)$$

where GT denotes the ground truth value.

### 4.2.4 Global Translation Tracking

The global translation tracking in this work is powered by velocity output of Motion Estimator and SMPL kinematic model. Specifically, we first compute the estimated velocity of the four joint endpoints (left and right forearms and lower legs) equipped with ToF-Inertial nodes and convert into pseudo stationary label $q_s$:

$$q_s^{(i)}(t) = \begin{cases} 1 & \text{if } ||v^{(i)}(t)||_2 < \epsilon \\ 1 - \frac{||v^{(i)}(t)||_2 - \epsilon}{0.2} & \text{if } \epsilon \leq ||v^{(i)}(t)||_2 < \epsilon + 0.2 \\ 0 & \text{otherwise} \end{cases} \quad (8)$$

Where $i$ denote endpoint index, the $\epsilon$ is a cut-off threshold to filter small jitter in $||v^{(i)}(t)||_2$ (we use $\epsilon = 0.05m/s$ in this work). Subsequently, based on $\varphi(t)$ provided by the Motion Estimator, we calculate the root translation relative to the endpoints is stationary, denoted as $s_{FK}^{(i)}(t)$, using the forward kinematics:

$$s_{FK}^{(1,2,3,4)}(t) = FK(\varphi(t - \Delta t)) - FK(\varphi(t)) \quad (9)$$

Where $\Delta t$ is time gap of 2 continuous captures. Then we define FK-based translation as follow:

$$s_{FK}(t) = \frac{\sum_{i=1}^4 q_s^{(i)}(t) \cdot s_{FK}^{(i)}(t)}{\sum_{i=1}^4 q_s^{(i)}(t)} \quad (10)$$

Similar to previous works[36], we fusion the $s_{FK}(t)$ with root velocity provide by Motion Estimator to obtain the final translation estimation:

$$\begin{aligned} s_{NN}(t) &= v_{root}(t - \Delta t) \cdot \Delta t \\ s(t) &= (1 - q_m) \cdot s_{FK}(t) + q_m \cdot s_{NN}(t) \end{aligned} \quad (11)$$

Where $q_m = \mathbf{min}(q_s^{(1)}, ..., q_s^{(4)})$ denotes the pseudo label of full-body moving (e.g., jumping on the air, sliding), and $s_{NN}$ denotes translation estimation base on neural network (Motion Estimators).

## 5 Experiment

### 5.1 Experimental Setup

**Synthetic Dataset.** We leverage the AMASS dataset [19] to synthesis a large-scale paired ToF-IMU-Motion data for motion estimators pre-training, which includes both IMU and ToF data simulation.

- **IMU Data Simulation**: Similar to previous works [8, 36, 11, 40], the IMU orientation and acceleration are calculated based on the global joint orientation of the SMPL [18] model and the trajectory of the selected mesh vertices.
- **ToF Data Simulation**: ToF data simulation is implemented using Unity. We simulate the ToF sensor with virtual depth cameras positioned on the rendered SMPL body, aligned with the hardware wearing setup. The original depth maps are then down-sampled into $4 \times 4$ to fit the configuration of ToF. More detailed settings are provided in the supplementary materials.

Table 1: Comparison of methods on DIP and our ToF-IP-DB datasets across multiple error metrics. We evaluated our ToF-IP on DIP with additional ToF synthesis.

| Method | ToF-IP-DB | | | | | DIP (with synthesis ToF) | | | | |
|---|---|---|---|---|---|---|---|---|---|---|
| | SIP Err | Ang Err | Pos Err | EndPos Err | Jitter | SIP Err | Ang Err | Pos Err | EndPos Err | Jitter |
| Transpose | 20.73 | 14.51 | 7.58 | 13.58 | 0.18 | 17.06 | 8.86 | 6.03 | 8.73 | 1.11 |
| TIP | 20.37 | 14.58 | 7.63 | 13.97 | 0.17 | 16.90 | 9.07 | 5.63 | 8.27 | 1.56 |
| PIP | 20.22 | 13.85 | 7.32 | 12.77 | 0.12 | 15.33 | 8.78 | 5.12 | 7.78 | 0.17 |
| DynaIP | 19.04 | 13.33 | 7.26 | 13.05 | 0.16 | 13.78 | 7.07 | 4.98 | 7.44 | 0.18 |
| PNP | 18.52 | 13.23 | 6.86 | 12.39 | 0.12 | 13.71 | 8.75 | 4.97 | 7.49 | 0.17 |
| ToF-IP(Ours) | **17.26** | **12.09** | **6.31** | **11.41** | 0.12 | **13.62** | **6.75** | **4.59** | **6.65** | 0.17 |

**ToF-IP-DB Dataset.** We collected a full-body motion capture dataset, containing over 20 types of movements, 208 minutes (749,000 frames) from 10 participants (3 male and 7 female) with heights ranging from 170 cm to 185 cm. All participants were informed about the purpose of the experiment and signed consent agreements. Participants were required to perform the following steps for data collection:

- Simultaneously wears an optical motion capture suit (with multiple optical markers attached) along with our 6-node ToF-Inertial motion capture prototypes.
- Performs a T-pose for IMU calibration.

During data collection, the participant is asked to perform diverse types of motion, e.g., dances, aerobics, and daily social activities. Sensor data and motion data are collected synchronously at 60Hz. The motion data is captured using the NOKOV marker-based optical motion capture system, including full-body pose and global translation. Each collection session lasts 6–10 minutes.

**Training Settings.** All our experiments run on a PC with an Intel(R) Core(TM) i7-13700KF CPU and an NVIDIA RTX 4080 GPU. The model is implemented using PyTorch 1.12.1 with CUDA 11.3. We use the Adam optimizer with a learning rate of $lr = 1 \times 10^{-3}$ and weight decay of $lr = 1 \times 10^{-6}$ during $n$ epochs training. The batch size was set to 512.

**Metrics.** We use the following five error metrics to evaluate the accuracy and quality of captured motion: *1) Angular Error* (°), which represents the global rotation error of all joints; *2) Positional Error* (*cm*), which is the joint position error of all joints; *3) SIP Error* (°), defined as the global rotation error of hips and shoulders; *4) Endpoints Positional Error* (*cm*), positional errors of the four ToF sensor attached joints (the left and right wrist and ankle); *5) Jitter* ($km/s^3$), denoting the jerk (time derivative of acceleration) of all body joints in the global space.

## 5.2 Comparison with SOTAs

**Quantitative Results.** Table 1 shows the evaluation results on the ToF-IP-DB and DIP [8] dataset. The results demonstrate that our method consistently outperforms existing approaches across all metrics, particularly in SIP angular error and positional error. The reduction in SIP error signifies more accurate tracking of upper arm movements and knee lifts, which is attributed to the ToF-enabled improvement in joint position estimation as we expected.

**Qualitative Results.** As illustrated in Fig. 4, we selected Tai Chi and Baduanjin movements from the ToF-IP-DB dataset, which are characterized by long durations and gentle velocities—conditions where inertial-only position tracking inherently fails to measure accurate joint position. In contrast, our method leverages ToF-derived direct distance measurements and introduce inter-joint distance constraints, leading to marked improvements in overall pose estimation accuracy.

## 5.3 Ablation Study

**Effectiveness of ToF Integration.** As shown in Table. 2, removing ToF leads to a noticeable decrease in all metrics, specifically in terms of EndPos Error, this validates our core motivation that the direct distance measurements from ToF sensors can improve the estimation accuracy of joint endpoints, thereby enhancing the estimation of non-sensor-attached measured joints (lower SIP). Notably, introducing ToF without using the proposed Node-centric Data Integration (NCI)

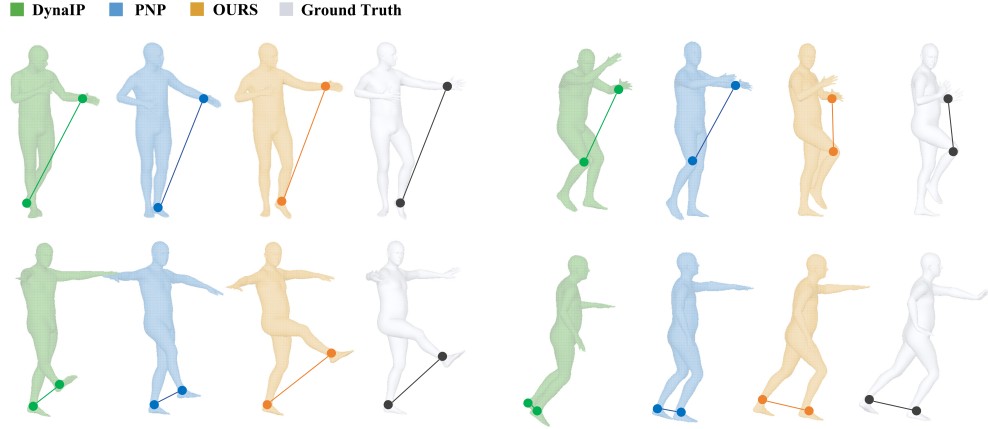

Figure 4: Qualitative comparisons with the state-of-the-art methods on our ToF-IP-DB dataset. We highlight the joint-to-joint distances in the ToF's line-of-sight direction.

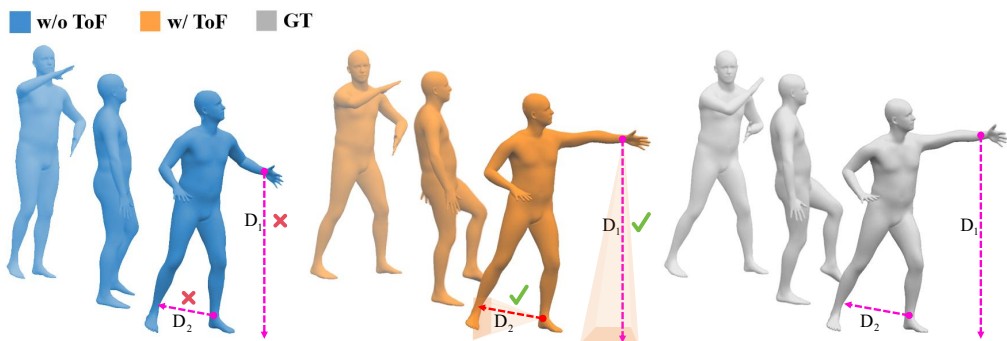

Figure 5: Qualitative comparison of results with and without ToF direct distance measurements. The pose estimation samples are from Case 1 and Case 4 in Table 2.

fails to achieve the desired improvement (Case 1 vs. Case 2), demonstrating the necessity of the proposed NCI for our ToF-Inertial motion capture framework. Qualitative results in Fig. 5 demonstrate how ToF integration significantly enhances inertial-only motion capture. The direct distance measurements provided by ToF effectively reduce the estimation errors of joint positions, thereby ensuring continuous and accurate pose estimation.

Table 2: Ablation study results on DIP and our ToF-IP-DB datasets (ToF integration).

| Case | ToF | NCI | ToF-IP-DB | | | | | DIP (with synthesis ToF) | | | | |
|---|---|---|---|---|---|---|---|---|---|---|---|---|
| | | | SIP Err | Ang Err | Pos Err | EndPos Err | Jitter | SIP Err | Ang Err | Pos Err | EndPos Err | Jitter |
| 1 | ✗ | ✗ | 18.87 | 13.14 | 6.88 | 12.27 | 0.07 | 16.34 | 7.64 | 5.80 | 8.49 | 0.13 |
| 2 | ✓ | ✗ | 18.95 | 12.89 | 6.92 | 12.39 | 0.13 | 16.10 | 7.50 | 5.47 | 8.08 | 0.17 |
| 3 | ✗ | ✓ | 17.90 | 13.18 | 6.81 | 12.06 | 0.13 | 15.52 | 7.30 | 5.30 | 7.79 | 0.17 |
| 4 | ✓ | ✓ | **17.26** | **12.09** | **6.31** | **11.41** | 0.12 | **13.62** | **6.75** | **4.59** | **6.65** | 0.17 |

**Effectiveness of Dynamic Spatial Positional Encoding.** As shown in Table 3, our proposed Dynamic Spatial Positional Encoding consistently outperforms both the traditional static encoding and its learnable variant (where parameters are optimized during training but remain fixed during inference) [2], demonstrating the effectiveness of modeling position in positional encoding for ToF-Inertial motion capture, Fig.6 further supporting the dynamic nature of our encoding scheme.

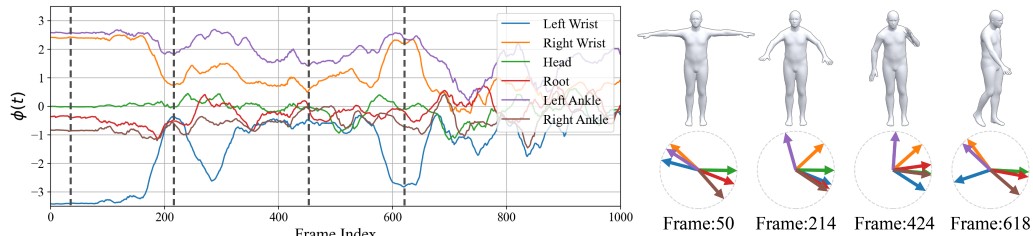

Figure 6: Quantitative visualization of the temporal dynamics of $\phi_n(t)$ across different motion types. Each curve represents the phase-based positional encoding of a specific sensing node over time.

Table 3: Ablation study results on DIP and our ToF-IP-DB datasets (positional encoding).

| Positional Encoding | ToF-IP-DB | | | | | DIP | | | | |
|---|---|---|---|---|---|---|---|---|---|---|
| | SIP Err | Ang Err | Pos Err | EndPos Err | Jitter | SIP Err | Ang Err | Pos Err | EndPos Err | Jitter |
| Static | 17.76 | 12.52 | 7.60 | 11.98 | 0.13 | 14.20 | 6.87 | 4.77 | 6.95 | 0.17 |
| Static (Learnable) | 17.56 | 12.20 | 7.36 | 11.56 | 0.14 | 13.95 | 6.85 | 4.62 | 6.73 | 0.18 |
| Dynamic Spatial (Ours) | **17.26** | **12.09** | **6.31** | **11.41** | **0.12** | **13.62** | **6.75** | **4.59** | **6.65** | **0.17** |

# 6 Limitations

Despite the superiority of our approach, several limitations highlight avenues for future research. The performance of our method is contingent on the availability and reliability of ToF-based distance measurements, which may be compromised in scenarios with occlusions, or limited field of view. Additionally, the inherent noise in ToF sensors introduces jitter in pose estimation, as reflected in our results. Furthermore, the current evaluation is conducted in controlled environments, and the generalization of our approach to more diverse and dynamic real-world scenarios remains to be validated. Future work could focus on improving robustness to ToF sensor limitations through hybrid models, reducing noise with advanced filtering techniques.

# 7 Conclusion

We introduced ToF-IP, a novel ToF-inertial motion capture system that overcomes the inherent limitations of sparse IMUs by integrating direct distance measurements from lightweight, body-mounted ToF sensors. Through a hardware-efficient design and a unified Transformer-based framework, ToF-IP achieves accurate joint position estimation while preserving the portability and wearability of existing IMU systems. Our software contributions, including Node-Centric Data Integration and Dynamic Spatial Positional Encoding, enable structured multi-sensor integration and dynamic spatial awareness, which are critical for handling complex, low-velocity, and non-linear human motions. Extensive experiments validate ToF-IP's effectiveness across diverse movement scenarios, setting a new standard for hybrid sensor-based human motion tracking.

# 8 Acknowledgments

The authors would like to thank Qiannan Cao, Yingqi Yang, Hui Zou, Lishuang Zhan, Jiabao Gan, Yile Pan, Ziyi Shan, Ran Li, Bingling Liu, Wenjing Wu, Jiaqi Li, Yuenan Ji, Ziqian Huang and Shuyang Xing for their help on live demos and dataset collection. This work was supported by the National Natural Science Foundation of China (Nos. 62472364, 62072383), the Public Technology Service Platform Project of Xiamen City (No. 3502Z20231043), the Xiaomi Young Talents Program / Xiaomi Foundation, and the Fundamental Research Funds for the Central Universities (No. 20720240058, "Young Eagle Plan" Top Talents of Fujian Province). This work was also supported by the National Key R&D Program of China (No. 2023YFC3305600).

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
