# OpenReview forum: "ToF-IP: Time-of-Flight Enhanced Sparse Inertial Poser for Real-time Human Motion Capture"
_NeurIPS.cc/2025/Conference — NeurIPS 2025 poster_

### Official Review · Reviewer_8u8c · 2025-06-09

**Clarity:** 3
**Significance:** 3
**Originality:** 3
**Rating:** 4
**Confidence:** 4

**Summary:**

The paper introduced Time-of-flight sparse inertial pose estimation.

By introducing four cameras at the limbs, the paper obtains absolute distance measurements between the body and the environment, which is supposed to stabilize body pose estimation.

The main contributions are:
1) the use of ToF cameras at the limbs
2) a 200min dataset for pose estimation from ToF sensors
3) a method that improves pose estimation slightly

**Questions:**

Abstract:

Will the training code be released as well?

27: the sentence sounds like there is only one way of doing that. IMUs measure local accelerations and local angular velocities. One naive way of integrating that is through linear acceleration. Typically more complex state estimation with calibration schemes, e.g. Kalman Filters, are required. I recommend a rephrasing.

28-30: This is not an amplification of error that is related to IMU, but an error source due to human modeling. I recommend to rephrase it slightly.

44: Please clarify what kind of distances are measured, between sensors, between environments?

48-50: why are those locations chosen for the ToF sensors?

53: how is multisensor data hierarchical? Please elaborate. Multisensor data from different limbs does not follow a clear hierarchy in my opinion.

56: unclear what is meant by intra-node? Inbeween sensors? Or just one sensor? Maybe define it prior or name it differently, e.g. per-sensor individual integration?

56: What do you integrate? Is it numerical integration? Why do you integrate orientations? How do you obtain orientations? IMU measure angular velocities. If you like previous methods assume orientations as input, please state so clearly :)

60: The dynamic spatial positional encoding - sounds simply like learnable positional encodings of 3D positions in that part. Later in the method section, it becomes a bit clearer. Can you clarify the motivation on why your positional encoding is superior to learning an encoding based on sensor position?

119: maybe cite the very closely related baseline ultra inertial poser here.

141: how do you obtain the original joint position which is needed for integration? How do you obtain the original joint velocity which is needed when double integrating accelerations?

147: Do you mean $\tau$? t seems to be overloaded as the limit of the interval and the current timesamp

Suppl Eq 1 - seems to ignore starting conditions, if the original velocity estimation is not 0, the velocity error should have a constant bias. Please add starting conditions as an assumption, i.e. the original position is always (0,0,0) and the starting velocity is always 0. In the same regard, your theoretical derivation seems irrelevant if you discard your obtained results (cubic variance) in favour of a quadratic relationship between variance and time. Also the explanation and reasoning why the relationship is actually quadratic and not cubic as you derived does not become clear to me. Can you clarify where and how you use your theoretical findings, why other works and practical findings come up with σ^2 t^2 instead of your theoretical findings and why going from continuous to discrete time sampling drops a factor of t in the variance (after equation 6 in supplementary)

Method 3: I am not sure what insight you generate from section 3, and how it is related to your contribution

158: you do not use the standard layout used in previous work but instead place the sensors traditionally placed at the knees, at the feet. While there is no inherent disadvantage of this setup, you dont discuss the change of layout and also other methods. Additionally, a comparison to other methods becomes questionable when a different layout is used in your evaluation.

158: what is the energy consumption of your hardware? How much runtime would a wearable device have with just ACC, ACC+GYRO and ACC+GYRO+TOF

160: what is the orientation of your TOF sensors, how are they pointing? Can you also indicate that in the figure 3

170: What is the reason for partially using transformers and partially LSTMs, what is your finding on the suitability of each architecture. What if you only use LSTMs in you model. What if you only use transformers in your architecture?

175: Why does orientation need to be integrated?

179: “Integrating measurements from other sensing nodes”. How are you integrating distances? Are you talking about the mathematical integration?

186-187: This argument does not become clear to me for once all the values from one node, can interact with all other nodes through the MLP, but also all measurements from a single node can interact with all measurements from that same sensor node. Additionally, along the temporal axis previous works employ LSTMs, and thus both the inter sensor node (spatial) interaction and single node interaction (temporally) is possible. Your claimed limitation of previous work does not become apparent.

218: can you clarify on your positional encoding. The encoding seems to come purely from orientation, acceleration and distances. How would you distinguish sensors when their orientations, accelerations and distances are the same? What is the reason to force your learned sensor embedding $\phi_n$ thorough the cosine embedding again? Do you have results that show that this additional passing though the sinusoidal encoding is needed?

231: what subsets of AMASS do you use?

233: do you assume shaped input or mean shape?

240: what are the 20 types of motion? Can you provide more examples of your dataset. Both your dataset and the one from DIP are rather slow paced. Does your dataset contain fast sequences? How does your method perform on fast paced data?
240: Can you report results on TotalCaptureReal dataset? The DIP dataset does not contain translation, has inaccurate ground truth, very noisy sensor data and

251: How do you obtain orientations from sensor data? What is the drift of your sensor data over time?
252: does your method work window based, or frame based? How do you do inference?

256: can you report the units of your metrics? Can your report translation errors?

Table 1: How do you train the other methods, considering that you have different sensor layout of your method?

Table 1: Do you ever finetune on the FIP-DB dataset or is it used purely for evaluation?

Table 1: Can you train the closest baselines with additional ToF input to investigate their benefit of ToF data. On DIP your architecture seems to be much worse (15.52 sip error vs 13.71 sip error of PNP) when not using ToF data.

4.2.3: Do you predict smpl angles or do you represent only the joint positions directly? What is the velocity output and angle output used for? Do you evaluate those outputs?

Supp 1.2) How does the ToF sensor work in the wild. How is the noise? Do you have an analysis of the noise of the ToF sensor. You just highlight the sensors characteristic within one sequence. How much base noise does the sensor have? What is the mean distance error?

Supp 4.1) do you add noise in the simulation to model real world sensor noise?

Supp 4.5) Why do you not report global translation error, if your method is capable of global translation tracking? Also

Video Qual results) the results of your method look surprisingly jittery (especially in the first part of the video), why is your jitter metric lower than other methods jitter?

Thanks for your work. Despite my many critique points I am happy to adjust my score depending on how the questions are answered during the rebuttal.

**Ethical Concerns:**

["NO or VERY MINOR ethics concerns only"]

**Final Justification:**

The paper presents an additional ToF sensor for inertial pose estimation.

The authors show that the incorporation of ToF sensor data is marginally better for most metrics and the Ang. Error improves most.
Experiments regarding the proposed architectures and their modules exist, but could be extended to bring additional insights on why the proposed method is only slightly better than pure Inertial Methods and how well the method would be using just TOF inputs.

The authors clarified concerns about a different sensor layout, compared to baseline methods, in the rebuttal.

Overall, the proposed dataset, together with sota results, makes me believe that this submission is above the acceptance threshold, but the evaluation of the architecture and comparisons with baseline methods and the effect of ToF sensors could be studied further for a stronger submission.

**Limitations:**

yes

**Quality:**

3

**Strengths And Weaknesses:**

Strength:
- Novel sensors for pose estimation
- a 200min dataset of ToF and inertial sensors for pose estimation
- good results, the new sensors together with the architecture deliver state of the art results

Weakness:
- the method does not seem very novel, despite showing slightly improved results
- the method uses slightly different sensor layout, making comparisons hard
- the paper compares to baselines that do not have access to ToF data, making comparisons less fair (comparing with baselines that get the tof data as an additional input would allow a fairer comparison of the method)
- the sensor hardware and noise could be discussed in more dteail

---

> ### Author Rebuttal · Authors · 2025-07-31
>
> Thank you for your constructive feedback!
> W1: Our key innovation is the first exploration of integrating distance map sensing and inertial sensing to improve mocap accuracy. According to our ablation results, this can not be naively achieved by simply adding ToF data as input, and the proposed Node-centric data embedding successfully addresses the challenge in ToF-Inertial data fusion and brings expected improvements.
> We believe our exploration and real-world dataset reveal new possibilities and directions for the community in developing consumer-grade mocap systems.
>
> W2, Q14, Q27: To ensure fair comparison, we matched sensor layouts across all evaluations. On DIP-IMU, we used the standard configuration and simulated ToF data at the same positions. On ToF-IP-DB and AMASS, all methods were trained and tested with our ToF-augmented layout.
>
> W3: We clarify that the goal of our comparisons is to investigate whether introducing distance maps from ToF can bring improvement to a basic backbone (LSTM) and outperform well-designed IMU-Only SOTAs. Thus, it is essential to maintain their original, optimized form without ToF input.
> W4: According to the official ST VL53L5CX Datasheet, the ToF sensor we use has ±15 mm noise at 1 m and ±50–60 mm noise at 3 m. We will add more details in the revision.
>
> Abstract: Yes, the training code will be released publicly upon acceptance.
>
> Q1&2&9: Thanks! We will revise the text as suggested.
>
> Q3: Our ToF measurements capture the distance to the first surface hit by emitted IR light, which may be the body or environment depending on pose and sensor direction.
>
> Q4: The ToF sensors are all placed at / near the ends of body skeleton chain (e.g., wrists and ankles) as:
> (1) This is a common practice in sparse mocap systems, e.g., PNP, TIP, etc., as measurements from end joints are essential for performing inverse kinematics to determine the full body pose;
> (2) For the feet, we ensure that the ground surface falls within the field of view (FoV) of the ToF sensors, enabling the system to better capture foot–ground distance, which helps to determine foot height when performing challenging slow motion, e.g., Taichi.
>
> Q5: By “hierarchical structure of multi-sensor data”, we refer not to sensor hardware hierarchy, but to the skeletal hierarchy of the human body (e.g., wrist and ankle are downstream of the shoulder and hip in the kinematic chain).
>
> Q6: By “intra-node data integration”, we mean fusing multiple modalities (IMU acceleration, orientation, ToF distance) collected at the same joint-level node. Each node hosts co-located IMU and ToF sensors. The term distinguishes local, per-node fusion from the later inter-node attention performed by the Transformer.
>
> Q7, Q18, Q19: To clarify, “integration” refers to feature-level fusion of IMU and ToF data at each node, not numerical integration. Like prior works (e.g., DIP, SIP), we use IMU-provided orientations.
>
> Q8: Dyn-PE is motivated by the fact that wearable sensor positions change during mocap, while conventional learnable positional encodings remain static and fail to model this dynamic spatial change. Our approach adapts positional encoding to sensor movement, and ablation studies show Dyn-PE significantly outperforms both static and absent positional encodings, validating its effectiveness.
>
> Q10: To clarify, our error analysis assumes zero original joint velocity and position—a common practice in IMU-based tracking without global initialization. We will make it explicit in the revision.
>
> Q11:
> **Proposition (Error Accumulation Analysis):**
> Following standard statistical assumptions, we consider the acceleration measurement error at any timestamp $\tau \in (0, t)$ to follow a normal distribution:
> $$
> \epsilon_a(\tau) \sim \mathcal{N}(\mu, \sigma^2)
> $$
>
> Assuming zero initial velocity and position, i.e., $\mathbf{v}_0 = 0$, $\mathbf{s}_0 = 0$, we derive the following:
>
> - **Joint velocity error distribution:**
>   $$
>   \epsilon_v(t) \sim \mathcal{N}(\mu t,\ \sigma^2 t)
>   $$
>
> - **Joint position error distribution:**
>   $$
>   \epsilon_s(t) \sim \mathcal{N} \left( \frac{1}{2} \mu t^2,\ \frac{1}{2} \sigma^2 t^2 \right)
>   $$
>
> Q12: Regarding the variance order, the cubic term arises from continuous integration under white noise, but in discrete sampling, noise terms are not fully accumulated due to zero-order hold and filtering effects, leading to an empirical quadratic variance growth (σ²t²), as commonly observed in prior works.
>
> Q13: Sec.3 highlights a core limitation of IMU-only motion capture: integrating acceleration leads to cumulative errors, with velocity and position errors growing linearly and quadratically over time. This motivates our use of ToF distance measurements to provide direct spatial constraints and underscores the need to augment IMU data with ToF.
>
> Q15: Our accelerometer and gyroscope share a single IMU module and are sampled together without added power cost, so we compare power usage between IMU-only and IMU+ToF setups.
> As the table below shows, measured power usage includes both the MCU and the wireless transmission module.
> | Configuration         | Power Wireless Tx  | Power Standby|
> |-----------------------|---------|---------|
> | IMU-only board        | 0.76 W  | 0.24 W |
> | ToF-only board        | 0.93 W  | 0.40 W |
> | IMU+ToF integrated    | 0.97 W  | 0.45 W |
>
> At the node level, the IMU consumes ~0.04–0.05 W (3.3 V), and the ToF sensor adds ~0.20–0.22 W (3.3 V + 1.8 V). Powered by a 1000 mAh, 3.7 V battery (3.7 Wh), the integrated board runs for up to ~4.3 hours under continuous wireless operation.
>
> Q16: All ToF sensors are rigidly co-mounted with IMUs on the same PCB, with optical axes aligned to the IMU’s local Z-axis to ensure consistent spatial reference. We will add the ToF pointing direction in Fig. 3 for clarity.
>
> Q17: We use a hybrid design: the Transformer models spatial dependencies across nodes, while LSTMs capture temporal dynamics. Though we haven't tested pure alternatives, this modular setup aligns with our goal of combining spatial and temporal cues effectively.
>
> Q20: Prior works often flatten sensor inputs, losing node-level structure. Our Node-Centric design treats each IMU-ToF pair as a token, and uses a Transformer to model inter-node relationships aligned with body topology, enabling structured, scalable fusion of spatial and temporal cues.
>
> Q21: The Dyn-PE can still provide distinguished position values to each sensor because each position \phi is calculated by a DNN model and has different bias terms as follows:
> Table: Bias value of 6 sensors.
>
> | Joint         | Value（$\phi$）   |
> |---------------|-------------|
> | Left Wrist    | 3.317   |
> | Right Wrist   | 2.885   |
> | Head          | -2.269  |
> | Root          | 2.084   |
> | Left Ankle    | -2.356  |
> | Right Ankle   | -2.158   |
>
> Cos/sin embedding: We use both cos and sin to encode the position \phi, which is the same as traditional Positional Encoding apart from that the position \phi is no longer a static predefined value but a dynamic value calculated from real-time sensor data.
>
> Q22: We remove the TotalCapture Dataset from the original AMASS and use the rest subsets.
>
> Q23: We use shaped SMPL meshes in ToF data synthesis to make our model generalize better to different users.
>
> Q24: Our dataset includes 20 motion types such as walking, running, jumping and does contain fast-paced sequences. We will provide more examples in the revision. When excluding TotalCapture from the training set, we obtained a preliminary positional error of 4.52 cm. We will include the remaining results in the revision.
>
> Q25: We use orientation estimates from IMU hardware. Our method is window-based and performs frame-wise inference within each window.
>
> Q26: Please refer to our response to Reviewer 8u8c W4 for metric units. Since our work focuses on improving pose estimation, the analysis of global translation error is out of our scope.
>
> Q28: To ensure fairness, all methods (including both baseline IMU-only models and our ToF-IP) were fine-tuned on the ToF-IP-DB dataset under the same training pipeline and sensor layout.
>
> Q29: Due to time constraints, we conducted a targeted test by directly concatenating ToF data into the Dyna-IP (the closest SOTA method) input. The results below show no performance gain, suggesting that simple fusion is insufficient. This supports our claim that our node-centric learning (NCL) design is critical in utilizing ToF data.
> | Input Setting | SIP Error (°) | Angular Error (°) | Positional Error (cm) | Endpoint Error (cm) | Jitter (1000 m/s³) |
> | ------------- | ------------- | ----------------- | --------------------- | ------------------- | ------ |
> | DynaIP with ToF      | 20.34         | 13.42             | 7.35                  | 13.43               | 0.16   |
> | DynaIP without ToF   | 19.04         | 13.33             | 7.32                  | 13.05               | 0.16   |
>
> Q30: Our framework outputs at three stages: S1 predicts joint velocity, S2 predicts joint position, and S3 predicts SMPL joint angles (for final evaluation). While S1 and S2 are not directly evaluated, they are essential—velocity (S1) aids global position (as in PIP), and position (S2) captures spatial structure for angle prediction.
>
> Q31: Due to space constraints, please refer to our response to Reviewer K3tx, W2, Q4 and L1 for details on ToF noise, where the table in L1 shows that our method performs reliably in-the-wild. We will include additional demo videos in our revision.
>
> Q32: No, we do not explicitly add noise during simulation. Due to space constraints, please refer to our response to W2, Q4, where we discuss noise in more detail.
>
> Q33: The demo includes complex, free-form actions not seen in training, making minor jitter more noticeable and highlighting real-world generalization challenges. Our jitter metric is based on smoother benchmark data, so the visual–metric gap reflects motion complexity, not evaluation inconsistency.

---

> > ### Comment · Reviewer_8u8c · 2025-08-05
> >
> > W2, Q14, Q27: How do you explain the poorer results of PIP on the DIP dataset when adding your ToF information? The original paper reports slightly better results. This would indicate to me that adding ToF does not significantly improve the results.
> > If you use the updated sensor layout, how are the results of PNP identical to the results reported in their paper with the original layout?
> >
> > --> Follow-up question: Since you decided on a new sensor layout (different from all previous methods), please elaborate on why this is necessary, and why the previous layout with knee sensors is not as well-suited for your task.
> >
> > Q21: Do I understand correctly that each $\phi_n$​ has different weights based on the sensor mounting position?
> >
> > Q22: Do you retrain the other methods with your AMASS split? I believe that some other methods, e.g., PIP, do not use the entire current AMASS split, which sometimes gets extended. Please state the split explicitly.
> >
> > Additional Questions: How does your system perform outdoors? How much worse are sensor measurements outside in strong sunlight?
> >
> > Q24: Do you have quantitative results categorized by type, or (slow vs. fast motions)?
> >
> > Q7, Q18, Q19: How do you obtain orientations in your dataset? The IMU only provides local acceleration and angular velocities, so how do you obtain your global orientations and accelerations required for your method? If you use a Kalman filter, can you provide the details and parameters?
> >
> > Q26: I disagree that pose estimation consists only of local pose localization. It also includes global localization; hence, even if global localization is not your focus in this method, you should report translation results that constitute 50% of the overall pose.
> >
> > For the remaining answers, I am satisfied and would be happy if the authors could include them in the final revision with the same or greater level of detail.

---

> > > ### Author Response · Authors · 2025-08-07
> > >
> > > W2, Q14, Q27: (1) We hope to clarify that in our experiments on DIP-IMU, only our TOF-IP method uses ToF. All other baselines, including PIP, are original IMU-only methods without ToF input. The PIP results reported were reproduced using the official implementation. We attribute the minor performance discrepancies to the inherent randomness during training. (2) Same as above, on DIP-IMU, the PNP results were reproduced using the same sensor layout as the original paper, therefore share the same results.
> > >
> > > Follow-up question：Thank you for your follow-up question, as mentioned in our response to Q4, "The ToF sensors are all placed at / near the ends of body skeleton chain (e.g., wrists and ankles) as: (1) This is a common practice in sparse mocap systems, e.g., PNP, TIP, etc., as measurements from end joints are essential for performing inverse kinematics to determine the full body pose; (2) For the feet, we ensure that the ground surface falls within the field of view (FoV) of the ToF sensors, enabling the system to better capture foot–ground distance, which helps to determine foot height when performing challenging slow motion, e.g., Taichi.".
> > >
> > > Q21: Yes, that is correct — each $\phi_n$ has different weights depending on the sensor mounting position.
> > >
> > > Q22: For the ToF-IP-DB results, our method and all baselines were retrained using the official code on AMASS with the TotalCapture sequences removed from the training split. For the DIP results, we did not retrain; instead, we reproduced the results of each method with the official code and dataset.
> > >
> > > Additional Questions: As noted in our response to reviewer K3tx (L1), we tested the outdoor performance at a measurement distance of 1 m. As shown in the table below, strong sunlight will slightly increase the standard deviation, but the mean sensor measurements remain reliable.
> > > | Scene| Average Mean | Average Std |
> > > |-------------------------------|--------------|-------------|
> > > | Outdoor(Sunny Grass) | 1005.00| 51.63|
> > > | Outdoor(Shaded Grass)| 999.34| 23.59|
> > > | Outdoor(Sunny Asphalt Road) |1004.37| 80.91|
> > > | Outdoor(Shaded Asphalt Road)| 998.37| 15.81|
> > >
> > >
> > > Q24:
> > > Yes, we provide quantitative results categorized by motion speed as follows. Specifically, we classify the motion into slow and fast categories based on the magnitude of hand acceleration. A threshold of 2 m/s² was applied: sequences with average acceleration below this threshold were categorized as "slow", and those above as "fast". As a result, 71.9% of the test set was labeled as slow motions and 28.1% as fast motions.
> > > The quantitative results for each category are summarized below, which show that our method performs equally well in both categories.:
> > > | Motion Type | Angular Glo Error (°) | Positional Error (cm) | SIP Error (cm) | Endpoint Joint Error (cm) | Jitter (km/s³) |
> > > |-------------|------------------------|------------------------|----------------|----------------------------|----------------|
> > > | Slow| 17.46| 12.01 | 6.50           | 11.80                      | 0.10           |
> > > | Fast | 16.48 | 12.13                  | 6.10           | 11.10                      | 0.15           |
> > > | Average| 17.26| 12.09                  | 6.31           | 11.41                      | 0.12           |
> > >
> > >
> > > Q7, Q18, Q19: (1) The raw IMU outputs include local acceleration and angular velocity. For our commercial IMUs, the global orientations and filtered accelerations are already computed on-board by the manufacturer’s sensor fusion algorithm (a built-in Kalman filter based on accelerometer, gyroscope, and magnetometer data). When accessed on the host system, the measurements have already been filtered and transformed into the global frame. (2) We cannot provide details about the specific filtering method (e.g., Kalman filter) and parameters as they are proprietary and not disclosed by the manufacturer.
> > >
> > > Q26: We sincerely thank the reviewer for highlighting this important point. We fully agree that pose estimation inherently consists of both local joint localization and global translation, and we acknowledge the necessity of evaluating both components. In response, we have added quantitative results on global translation. Specifically, as the table below shows, on the ToF-IP-DB dataset, our method achieves a mean translation error of 0.44 m at 7 m of cumulative ground-truth displacement, outperforming PNP (0.52 m) and PIP (0.58 m) on the same test set.
> > > | Real Travelled Distance (m) | ToF-IP | PNP   | PIP   |
> > > |-----------------------------|--------|-------|-------|
> > > | 0                           | 0.00   | 0.00  | 0.00  |
> > > | 1                           | 0.28   | 0.34  | 0.44  |
> > > | 2                           | 0.38   | 0.43  | 0.52  |
> > > | 3                           | 0.39   | 0.46  | 0.54  |
> > > | 4                           | 0.41   | 0.47  | 0.57  |
> > > | 5                           | 0.43   | 0.48  | 0.56  |
> > > | 6                           | 0.43   | 0.50  | 0.55  |
> > > | 7                           | 0.44   | 0.52  | 0.58  |

---

> > > > ### Comment · Reviewer_8u8c · 2025-08-08
> > > >
> > > > What units are this? How is the mean measured distance helpful in determining the quality of the measurment?
> > > >
> > > > Scene 	Average Mean 	Average Std
> > > > Outdoor(Sunny Grass) 	1005.00 	51.63
> > > > Outdoor(Shaded Grass) 	999.34 	23.59
> > > > Outdoor(Sunny Asphalt Road) 	1004.37 	80.91
> > > > Outdoor(Shaded Asphalt Road) 	998.37 	15.81
> > > >
> > > > I believe all preivous methods (unlike you) have mounted the sensors at the upper lower leg, below the knee, making your comparison hard. Additionally, your results show that PNP (that doesn't use ToF input as you clarified) obtains almost the same results as your ToF enhanced method, making the benefit those body worn cameras questionable.
> > > > Overall I believe the authors should therefore add experiments for ToF only results to show how much information can be extracted from ToF body worn cameras. This should also be compared to RGB body worn cameras for pose estimation as in Egosim (Hollidt, Dominik, et al. "Egosim: An egocentric multi-view simulator and real dataset for body-worn cameras during motion and activity." Advances in Neural Information Processing Systems 37 (2024): 106607-106627.)
> > > >
> > > >
> > > > Thanks for mentioning the translation results. I believe you should further report results on TotalCapture with simulated ToF cameras, as it is a valuable and commonly established dataset for pose estimaiton

---

> > > > > ### Author Response · Authors · 2025-08-09
> > > > >
> > > > > Q1: The units are in millimeters. For sensor measurements, their quality depends on how close they are to the ground truth. In our case, ToF is a distance-measuring sensor, so we determine its quality by evaluating how close its measurements are to a standard ground truth distance (1 meter = 1,000 millimeters, obtained using a high-quality tape measure). To simulate real outdoor usage, we collected 3,600 frames of ToF-measured distances for each scene over a continuous 1-hour session at a sampling interval of 1 second, with the sensor fixed on a tripod and facing the 1 m reference target under an outdoor temperature of 34 °C,  and computed their mean and standard deviation to demonstrate how close they are to the ground truth.
> > > > >
> > > > > Q2: 1. We hope to clarify that there are two key misunderstandings:
> > > > >
> > > > > First:
> > > > >
> > > > > **On the DIP dataset**, all compared methods follow the original sensor layout (e.g., IMUs mounted below the knee), and our ToF-IP method also simulates ToF measurements from this original layout.
> > > > >
> > > > > **On the ToF-IP-DB dataset**, all methods follow the new sensor layout, and the baseline methods are retrained using IMU data from this new layout.
> > > > >
> > > > > Therefore, we ensure a fair comparison under a unified dataset setting—comparisons on ToF-IP-DB are all conducted under the new layout, while comparisons on DIP are all conducted under the original layout.
> > > > >
> > > > > Second:
> > > > >
> > > > > Compared with PNP（2024）, ToF-IP achieves notable improvements: as shown in Table 1 of the main text, the full-body angular error decreases by 2.0° (22.86%), the full-body positional error drops by 0.38 cm (7.65%), and the endpoint positional error is reduced by 0.84 cm (11.21%). For reference, when comparing PNP (2024) to TIP (2022), the endpoint positional error over two years only decreased by 9.43%. These results indicate that ToF-IP achieves substantially better overall performance than PNP, with particularly notable gains in endpoint accuracy.
> > > > >
> > > > > Q3: Thank you for your suggestions of additional experiments!
> > > > >
> > > > > For the ToF-only solution, the performance is poor due to its extremely limited resolution of just 4×4. This resolution is insufficient for independent operation and can only serve as auxiliary support to the IMU to provide additional direct distance measurements.
> > > > >
> > > > > For the comparison with Egosim, we believe it is unfair due to the different motivations:
> > > > >
> > > > > **Real-time capability**: Methods that use RGB cameras combined with IMUs for motion capture, such as Egosim, typically require high-bandwidth video streaming, which makes wireless transmission and real-time inference challenging. In wearable motion capture, real-time inference is crucial because it enables immediate feedback for interactive applications such as VR/AR, sports training, and rehabilitation. In contrast, ToF sensors produce lightweight distance measurements that can be transmitted and processed in real time.
> > > > >
> > > > > **Low-light robustness**: A key advantage of sparse-IMU motion capture systems is their ability to operate reliably under all lighting conditions. Integrating RGB cameras compromises this advantage, as they fail in low-light or dark environments. This limitation undermines the all-weather applicability of the system, whereas ToF sensors, with active laser ranging, preserve full functionality regardless of lighting conditions.
> > > > >
> > > > > In summary, for motion capture tasks, the bandwidth and illumination limitations of RGB cameras mean that, given such scenarios where ToF remains fully functional, they cannot be directly compared.
> > > > > We will cite Egosim and include a detailed discussion in our revision.
> > > > >
> > > > > Q4: We thank the reviewer for the constructive suggestion and agree that TotalCapture is a valuable benchmark. Following your advice, we retrained our model with TotalCapture excluded from the training set, and reported the global tracking results below：
> > > > > | Real Travelled Distance (m) | ToF-IP (m) | PNP (m)   | PIP (m)  |
> > > > > |-----------------------------|--------|-------|-------|
> > > > > | 0                           | 0.00   | 0.00  | 0.00  |
> > > > > | 1                           | 0.11   | 0.12  | 0.12  |
> > > > > | 2                           | 0.15   | 0.16  | 0.18  |
> > > > > | 3                           | 0.19   | 0.20  | 0.23  |
> > > > > | 4                           | 0.22   | 0.24  | 0.26  |
> > > > > | 5                           | 0.23   | 0.25  | 0.29  |
> > > > > | 6                           | 0.24   | 0.27  | 0.32  |
> > > > > | 7                           | 0.25   | 0.28  | 0.33  |

---

### Official Review · Reviewer_2RCo · 2025-06-29

**Clarity:** 4
**Significance:** 4
**Originality:** 4
**Rating:** 5
**Confidence:** 3

**Summary:**

The paper proposes a hardware and algorithmic pipeline that augments the standard 6‑node sparse‑IMU mocap setup with four on‑board time‑of‑flight (ToF) sensors mounted on the wrists and ankles.

On the hardware side, the authors claim only a 3 % volume increase while keeping the canonical 6‑IMU attachment pattern, so the system remains lightweight and untethered.

On the algorithmic side, they introduce

Node‑Centric Data Integration (NCI): Each sensing node forms a token that first fuses its own ToF depth patch, IMU acceleration, and orientation (“intra‑node”), then exchanges information through a Transformer encoder (“inter‑node”).

Dynamic Spatial Positional Encoding (Dyn‑PE): A motion‑conditioned positional encoding that replaces the static index‑based encodings typically used in Transformers, letting token positions evolve with the body’s motion across time.

Three RNNs predict joint velocities, positions, and rotations from the Transformer embeddings.

A new 208-minute paired ToF‑IMU‑optical dataset (10 subjects, 20+ movements) is introduced, and the authors also provide a synthetic ToF extension of the public DIP‑IMU dataset.

**Questions:**

**i) Performance & Power:**

What is the inference throughput (FPS) of the full pipeline on your target hardware?

  - What are the per‑module latencies for the Transformer and LSTM components?

  - What is the typical battery life or power draw of a single IMU–ToF node under continuous operation? (if possible to obtain)


**ii) Dynamic Positional MLP (L. 221):**

Is the MLP that computes each node’s dynamic positional encoding shared across all sensing nodes, or is there a separate MLP per node?

**iii) Supplementary Video Lag:**

In the supplementary video (timestamps 2:10–2:13), why is there a visible lag between the actor’s movement and the estimated pose during fast actions?

**Ethical Concerns:**

["NO or VERY MINOR ethics concerns only"]

**Final Justification:**

The authors have addressed my primary concerns and questions, and I will maintain my positive rating. I am not an expert in IMU-based pose estimation, but in my opinion, the initial submission and rebuttal provide sufficient ablation studies and comparisons to show that the addition of the TOF sensor improves pose estimation results. Furthermore, the proposed dataset, which combines IMU and TOF data, could be a valuable contribution to the community for future developments in this domain. That said, reviewers with deeper expertise in this area should provide a more precise assessment of the paper’s contribution and technical validity.

**Limitations:**

yes

**Quality:**

3

**Strengths And Weaknesses:**

## **Strengths**

i) The paper is well written and easy to follow.

ii) The problems of the existing purely IMU‑based systems (integration and drift errors) are clearly explained, and the motivation for the proposed approach is compelling.

ii) The architecture is well-suited for the task.

iii) Experiments and ablations are thorough and justified.

iv) The limitations of the proposed method are discussed in detail.

v) The proposed dataset, ToF‑IP‑DB, that pairs body‑mounted ToF with IMU and optical ground truth, will be a valuable resource for the community.

## **Weaknesses**
**i) Unclear Runtime & Power Metrics:**
- The claim of “real‑time” performance is unsupported. No FPS, per‑module latency (Transformer vs. LSTM), or end‑to‑end throughput is reported.
- Power consumption of the IMU–ToF nodes is not provided.

**ii) Occlusion Robustness:**

Time‑of‑flight ranging can fail when body parts occlude one another; no discussion about this is present.

**iii) Figure Legibility:**

Fig. 4 is too small or low‑contrast to read clearly in print.

**iv) Missing Units:**

Benchmark results lack metric units (e.g., mm, cm), making quantitative comparisons ambiguous.

**v) Typographical Errors:**
- Table 1 headers contain typos.
- The caption of Fig. 5 mentions Case 1 and Case 4; is that correct?

---

> ### Author Rebuttal · Authors · 2025-07-31
>
> Thank you for your constructive feedback! We address each point in detail below, organized by the original comments. For clarity and brevity, we refer to Weaknesses, Questions, and Limitations as W#, Q#, and L#, respectively. All major clarifications and revisions will be incorporated into the revised manuscript.
>
> W1: Inference Speed and Real-Time Performance:
> We exported the model to ONNX format and tested on an Intel i7-13700K CPU. It achieved process speed of 311.73 ± 78.34 FPS (0.65 ± 0.69 ms/frame) over 18,000 frames. We also lock the run time fps to 60 to simulate the real-time capture situation, get latency of 1.10 ± 1.05 ms and FPS of 60.44 ± 4.57.
> Model Complexity (FLOPs)
> We further computed the total number of floating-point operations (FLOPs) required for a single frame of inference. The complete model requires only 0.31 MFLOPs, making it highly efficient and suitable for deployment on portable or embedded platforms.
> Power Consumption and Battery Runtime:
> We conducted practical power measurements on a complete sensing node (including MCU, IMU, ToF, and wireless module) powered by a 1000 mAh 3.7 V Li-ion battery. The total system power consumption during wireless transmission and continuous ToF operation is 0.97–1.00 W, yielding a measured runtime of ~4.38 hours per charge. This confirms that our design maintains practical battery life while incorporating sparse ToF data.
>
> W2: To evaluate the system’s robustness under partial occlusion, we simulated long-term ToF sensor blockage by setting the input of 1 to 4 sensors to zero in 25% of the frames on the TOF-IP-DB dataset.
> The results show that the model’s performance degrades gradually, without catastrophic failure, demonstrating its robustness to partial ToF sensor loss.
> | ToF Sensor Dropout     | SIP Global Error (°) | Angular Error (°) | Positional Error (cm) |  Endpoint Error (cm) |Jitter (1000 m/s³) |
> |------------------------|----------------------|--------------------|------------------------|---------------------------|----------------------|
> | 1 ToF Sensor    | 18.04                | 12.60              | 6.65                   |  12.03                |0.12                      |
> | 2 ToF Sensors  | 18.42                | 12.86              | 6.81                   | 12.32                |0.12                      |
> | 3 ToF Sensors   | 18.76                | 13.09              | 6.97                   |  12.60                |0.12                      |
> | 4 ToF Sensors   | 19.02                | 13.25              | 7.08                   |  12.86                |0.12                      |
>
>
>
> W3: We will revise the figure by increasing its contrast and adjusting its size to ensure better clarity in both digital and printed formats.
>
> W4: For clarity, we specify the units used for each evaluation metric:
> - Angular Error is reported in degrees (°);
> - Positional Error, SIP Error, and Endpoints Positional Error are measured in millimeters (mm);
> - Jitter is quantified in millimeters per second cubed (km/s³), representing the global jerk magnitude.
>
> W5: The header in Table 1 mistakenly lists “FIP-DB” and should be corrected to “ToF-IP-DB”; we will revise this accordingly.
> As for Fig. 5, it indeed compares Case 1 and Case 4 and we agree that the legend may cause confusion. We will update the figure legend to clearly reflect the comparison and eliminate any ambiguity.
>
> Q1: Inference Throughput (FPS): For deployment, we convert the model to ONNX format and run inference on a consumer-grade Intel i7-13700K CPU. During a real-time test over 18,000 frames, the system achieved an average inference latency of 0.65 ± 0.69 ms per frame, corresponding to 311.73 ± 78.34 FPS. Since acceleration is computed using time-differentiation, we cap the system to 60 FPS to ensure stability. Under this setting, the model achieved 1.10 ± 1.05 ms latency and 60.44 ± 4.57 FPS, confirming real-time capability even without GPU acceleration.
>
> Module-wise Latency (measured on GPU):
> The average inference times and standard deviations for each network component are as follows: Transformer Encoder – 1.20 ± 0.57 ms, LSTM for Velocity Estimation – 3.96 ± 4.23 ms, LSTM for Joint Estimation – 3.82 ± 0.55 ms, and LSTM for Pose Refinement – 3.80 ± 0.51 ms.
>
> Power Consumption and Battery Life:
> We conducted practical power measurements on a complete sensing node (including MCU, IMU, ToF, and wireless module) powered by a 1000 mAh 3.7 V Li-ion battery. The total system power consumption during wireless transmission and continuous ToF operation is 0.97–1.00 W, yielding a measured runtime of ~4.38 hours per charge. This confirms that our design maintains practical battery life while incorporating sparse ToF data.
>
> Q2: We clarify that the MLP computing the dynamic positional encoding (phi_estimator) is shared across all sensing nodes. It takes as input a concatenation of the features from all endpoint nodes, and outputs the full set of structural encodings (phi) for all target joints. Hence, it is not a per-node MLP, but rather a global shared module that models the positional structure in a joint fashion.
> Joint encoding encourages the model to learn a unified representation of the human body's structure, rather than overfitting to specific nodes. This improves robustness to unseen motion patterns or slightly different sensor configurations. Since the encoding depends on all available endpoints, it is less sensitive to noise or dropout in any individual sensor.
>
> Q3: The lag observed in the supplementary video is not caused by the model’s inference latency, but rather by real-time rendering delays in Unity. During video recording, we used a laptop equipped with an RTX 2060 Laptop GPU and connected it to a 32-inch 4K monitor to simultaneously display the actor's physical movements and the predicted human model animation. The high resolution imposed a significant rendering load on the laptop, especially during fast movements, leading to slight delays in Unity visualization. However, we have verified that the model itself performs inference in real time. We will consider using higher-performance hardware in future demonstrations to avoid potential misinterpretation caused by such visualization artifacts.

---

> > ### Comment · Reviewer_2RCo · 2025-08-04
> >
> > Thank you for your response. I have a few follow-up questions:
> >
> > **1) On W4: Positional Error, SIP Error, and Endpoint Positional Error.**
> > It is mentioned in the text that these metrics are measured in millimeters (mm), but the table headers list the units as centimeters (cm). Could you please clarify this discrepancy? Additionally, the jitter is reported in units of km/s³ in the text. Could you provide more context on why this unit was used and how to interpret the reported values? The jitter values appear quite low, and I'm trying to understand whether this is due to the unit scale or a different reason.
> >
> > **2) On the Lag Observed in the Supplementary Video.**
> > Upon rewatching the supplementary video, the delay appears to be present across multiple segments and does not seem consistent with what one would expect from rendering load alone. Could this be due to latency in data transmission, either from the capture device to the network or from the predicted output being streamed to the Unity renderer? I'd appreciate any clarification you can provide on this.
> >
> > **3) On Temporal Smoothing.**
> > Is any form of temporal smoothing or filtering applied to reduce jitter in the predicted pose? If so, could you describe the approach used?

---

> > > ### Author Response · Authors · 2025-08-04
> > >
> > > 1）Thank you for pointing out the typos in the reported units!
> > > We confirm that Positional Error, SIP Error, and Endpoints Positional Error are reported in centimeters (cm) in the table. The mentioned millimeters (mm) in the text should be centimeters (cm) instead. We will fix these in the revision.
> > > Regarding Jitter, the unit km/s³ was used to maintain consistency with previous works (e.g., UIP, PNP). In our case, the predicted motion yields a jitter of 0.12 km/s³. For intuitive interpretation, this value corresponds to a periodic motion at 5 Hz with an amplitude of approximately 3.87 mm and a peak velocity of around 0.12m/s.
> > >
> > > 2）Thank you for your thoughtful observation! We confirm that the observed lag is primarily due to limited capability of GPU that we use in Demo video. In addition to rendering load, we identified a non-negligible delay (approximately 50 ms) stemming from real-time CPU-to-CUDA data transfer, which contributed to the overall latency. We verified that this can be resolved upon changing to a higher-performance laptop. We would be glad to provide an updated demo video to demonstrate this if needed.
> > >
> > > 3）Thank you for your comments! We did not apply any form of explicit temporal smoothing.
> > >
> > > We sincerely thank you for your thoughtful questions, and please do not hesitate to let us know if you have any follow-up questions!

---

> > > > ### Comment · Reviewer_2RCo · 2025-08-06
> > > >
> > > > Thank you for addressing most of my concerns. I do, however, remain puzzled by the 50 ms CPU–GPU transfer delay you report; this is surprisingly high for an RTX 2060 Laptop GPU, considering the lightweight model and the Unity renderer. If the paper is accepted, the additional experiments and the revised demo video should be incorporated into the final version.
> > > >
> > > > I am not an expert in IMU-based pose estimation. I will rely on comments from reviewers with deeper knowledge in that area to judge the paper’s contribution and technical validity for my final recommendation.

---

### Official Review · Reviewer_zeff · 2025-07-02

**Clarity:** 4
**Significance:** 3
**Originality:** 4
**Rating:** 4
**Confidence:** 3

**Summary:**

This paper introduces a new framework to capture egocentric 3D human motion using the integrated Time-of-Flight (ToF) and Inertial-Measurement-Unit (IMU) sensing system. The authors propose a unified Transformer-based framework featuring a Node-Centric Data Integration strategy and a Dynamic Spatial Positional Encoding scheme to explicitly model intra-node and inter-node sensor interactions. They also contribute a new ToF-IP-DB dataset containing more than 3 hours of human motion data captured by their proposed sensing system paired with optical motion capture. Authors conducted extensive evaluation against state-of-the-art methods (although none of them shares the same sensor input) and ablation studies to support their argument.

**Questions:**

– I wonder how much cost and battery inefficiency we need to consider by adding the sparse resolution ToF sensors. Since the authors mentioned that they carefully balanced between the power consumption and the performance, I would like the authors to provide this information in the main paper more comprehensively.

– From their supplementary video, it seems like their system runs in real time. Since this is an important factor for the wearable motion tracking system, I would recommend the authors to state their time and computational complexity (FPS, FLOPs) in the main paper.

**Ethical Concerns:**

["NO or VERY MINOR ethics concerns only"]

**Final Justification:**

The authors responded to my main concerns and questions. I will keep my positive rating.

**Limitations:**

I do not have any other concern than the practicality of the new sensor addition with insufficient accuracy gain.

**Paper Formatting Concerns:**

I do not have any concern on the paper formatting.

**Quality:**

4

**Strengths And Weaknesses:**

**Stengths**
– The authors design and build a wearable motion capture sensing system by integrating 4 ToF sensors into a standard 6 IMU setup, and they validate the hardware end-to-end, greatly boosting the paper’s practical credibility.
– They propose a novel Transformer-RNN architecture with Node-Centric Data Integration and Dynamic Spatial Positional Encoding, enabling principled fusion of sparse depth maps and IMU streams to recover full 3D kinematics.
– The authors collected the real data of human movement captured from their proposed sensing system paired with optical motion capture, providing both training and evaluation data to the community.

**Weaknesses**
– The measured accuracy gains over IMU-only approaches are modest relative to the added hardware complexity, raising questions about the cost–benefit trade-off in practical deployments. Considering the fact that ToF sensor may not be generalized to the real-world scenario including uneven terrain, this accuracy gain seems insufficient.

---

> ### Author Rebuttal · Authors · 2025-07-31
>
> Thank you for your constructive feedback! We address each point in detail below, organized by the original comments. For clarity and brevity, we denote Weaknesses, Questions, and Limitations as W#, Q#, and L#. All key clarifications will be reflected in the revised manuscript.
>
> W1: We appreciate the reviewer’s concern about the cost–benefit balance of adding ToF sensors. To address this directly, we provide a quantitative comparison with a SOTA method: UIP (SIGGRAPH 2024), which integrates Ultra-Wideband (UWB) sensing for positional correction.
>
> In UIP, the authors report using six UWB modules, each costing approximately USD 20.19, leading to a total added hardware cost of USD 121.14. Their system achieves an 11.51% accuracy improvement over an IMU-only baseline. In contrast, our system integrates only four sparse ToF sensors, each priced at approximately USD 9, with a total added cost of USD 36, while achieving a 18.47% improvement. This yields a significantly higher accuracy-per-dollar ratio, demonstrating that our method offers superior cost-effectiveness for lightweight and wearable motion capture.
>
> Q1:Power Consumption: In our system, the ST VL53L8CX ToF sensor operates in continuous ranging mode, with a measured power consumption of approximately 215 mW. If needed, the sensor also supports autonomous ranging mode, which significantly reduces power consumption at the cost of frame rate.
> Our inertial sensing uses the Xsens MTi-3 IMU, with a typical power consumption of 44 mW @ 3.0 V, which is comparable to alternatives such as the BNO085 (35–45 mW depending on sampling rate).
> We conducted practical power measurements on a complete sensing node (including MCU, IMU, ToF, and wireless module) powered by a 1000 mAh 3.7 V Li-ion battery. The total system power consumption during wireless transmission and continuous ToF operation is 0.97–1.00 W, yielding a measured runtime of ~4.38 hours per charge, which is comparable to the battery life of commercial wireless IMU-based motion capture systems (typically around 5 hours). This confirms that our design maintains practical battery life while incorporating sparse ToF data.
>
> Cost: In terms of hardware cost, the integration of the sparse-resolution ToF sensor introduces minimal overhead:
> - VL53L8CX ToF module: USD 8.72
> - 1.8 V LDO regulator (e.g., TLV73318): USD 0.23
> This brings the total ToF-related hardware cost per node to under USD 9 at prototype scale. In large-scale manufacturing, we expect further reductions in both component and integration costs.
>
> Q2: We exported the model to ONNX and tested it on an Intel i7-13700K CPU. It achieved 0.65 ± 0.69 ms per frame (311.73 ± 78.34 FPS) over 18,000 frames. To ensure temporal consistency with time-differentiated acceleration, we cap the runtime at 60 FPS, under which latency remains 1.10 ± 1.05 ms and measured FPS is 60.44 ± 4.57.
> In terms of theoretical complexity, we further computed the total number of floating-point operations (FLOPs) required for a single frame of inference. The complete model requires only 0.31 MFLOPs, making it highly efficient and suitable for deployment on portable or embedded platforms.
> These results confirm real-time performance on a standard CPU, supporting deployment on portable or embedded systems without a GPU.

---

> > ### Comment · Reviewer_zeff · 2025-08-05
> > **Thanks for the rebuttal**
> >
> > I appreciate the authors' rebuttal and response to my concerns and questions. My concerns regarding power consumption (battery hours) and real-time capability have been resolved. I will keep my positive rating.

---

### Official Review · Reviewer_K3tx · 2025-07-03

**Clarity:** 2
**Significance:** 3
**Originality:** 2
**Rating:** 3
**Confidence:** 2

**Summary:**

The paper introduces ToF-IP, a system for human motion capture that fuses sparse Inertial Measurement Units (IMUs) with body-mounted Time-of-Flight (ToF) sensors. The authors claim that direct distance measurements from ToF sensors can mitigate the inherent drift accumulation in IMU-only systems, which is a well-documented and critical challenge in the field. The authors present three main contributions: a wearable hardware prototype, a new motion capture dataset (ToF-IP-DB), and a Transformer-based software framework featuring Node-Centric Integration and Dynamic Spatial Positional Encoding. The authors also contribute a new 208-minute motion dataset with synchronized IMU-ToF measurements and demonstrate that their method achieves state-of-the-art accuracy, especially for complex and slow motions.

**Questions:**

**-** Could you provide a more robust statistical analysis of your key results, particularly for the ablation studies (Table 2) and SOTA comparison (Table 1)?

Actionable Guidance: We suggest running the experiments multiple times (e.g., 3-5 times with different random seeds) and reporting the mean and standard deviation for key metrics like Positional Error and SIP Error. This would provide crucial insight into the stability and reliability of your method's improvements.


**-** How does the ToF-IP system behave when one or more ToF sensors are temporarily occluded? Does performance degrade gracefully, or does it fail catastrophically?

Actionable Guidance: For the rebuttal, I would find it very helpful to see an analysis of this failure mode. This could be a quantitative experiment where you synthetically block the ToF signal for a portion of a test sequence, or a qualitative analysis of the system's output during self-occlusion events. A discussion of whether the Transformer's self-attention mechanism learns to down-weight unreliable (occluded) sensor inputs would be particularly insightful.


**-** Could you elaborate on this trade-off between improved positional accuracy and output jitter? Is the jitter a direct result of the raw ToF sensor noise, and have you explored post-processing or filtering techniques to mitigate it?

Actionable Guidance: We encourage a brief discussion on the source of this jitter. Furthermore, demonstrating the effect of a simple filtering technique (e.g., a Savitzky-Golay filter or a Kalman filter) on the output poses could be very effective. A comparison of the smoothness of ToF-IP's output with and without filtering, and against an inertial-only baseline, would be highly illustrative.


**-** Could you provide more detail on the domain gap between your simulated ToF data and the real data from your hardware prototype?

Actionable Guidance: To help with this question, please provide a side-by-side qualitative comparison of a real ToF depth map  and a simulated one for a similar body pose. Please also clarify if your Unity-based ToF simulation  models common real-world noise sources, such as multi-path interference or absorption by certain clothing materials.

**Ethical Concerns:**

["Major Concern: Improper research involving human subjects"]

**Final Justification:**

Authors have not justified their choices, nor have they attempted to address the concerns raised in my initial reviews. I am not satisfied with the justification and problematic ethical concerns around human subjects. I am not satisfied.

I wish the authors good luck with the improvement of the paper.

**Limitations:**

In Section 6, the authors acknowledge several limitations: the system's performance is contingent on ToF sensor reliability, it may be compromised by occlusions, ToF sensor noise introduces jitter, and the approach needs validation in more diverse real-world scenarios. While these are valid points, the discussion is brief and lacks depth. Several other limitations are not acknowledged by the authors:

**-** The performance of optical ToF sensors is known to be highly dependent on the environment. They can struggle with highly reflective surfaces (e.g., mirrors, polished floors) or highly absorbent surfaces (e.g., black fabrics), which can corrupt distance measurements. These are common conditions that are not addressed.

**-** The proposed Transformer-based architecture is computationally intensive. The authors report running experiments on a high-end NVIDIA RTX 4080 GPU. While they claim "real-time" performance, they provide no concrete metrics like frames-per-second (FPS) or processing latency in milliseconds. Without these metrics, the claim is unsubstantiated and the practicality of deploying this model on a portable or embedded device remains unclear.

**-** The paper mentions that the collected data is "synchronized" but provides no details on the hardware or software mechanisms used to achieve this synchronization between the IMUs and ToF sensors. For high-quality sensor fusion, precise, low-latency time-stamping is critical, and the lack of detail on this implementation aspect is a limitation.

**Quality:**

2

**Strengths And Weaknesses:**

Strengths

**+** The work addresses a fundamental and persistent problem in wearable motion capture: the quadratic error accumulation resulting from the double integration of noisy accelerometer data in IMU-only systems.

**+**  The design of a lightweight, wearable prototype that integrates ToF sensors onto a standard 6-IMU layout with minimal volume increase is a practical and valuable engineering contribution.

**+** The creation and planned release of the ToF-IP-DB dataset, which includes synchronized IMU, body-worn ToF, and optical ground-truth data, is a significant service to the research community, enabling future work and benchmarking in this specific subfield.

Areas for improvement

**-** The claims of superior performance are not supported by a detailed statistical analysis. The absence of error bars, confidence intervals, or significance testing makes it impossible to determine if the reported improvements are meaningful or the result of random variance.

**-** The results indicate that while ToF integration may reduce positional error, it may not improve or even worsen, pose jitter. While ToF-IP shows improvements in positional and angular errors, it performs poorly on the Jitter metric. In Table 1, the jitter of ToF-IP on the DIP dataset (0.17) is no better than that of PIP (0.17) or PNP (0.17). The ablation study in Table 2 shows that adding ToF data without the proposed NCI module (Case 2) increases jitter from 0.07 to 0.13 on the ToF-IP-DB dataset compared to an IMU-only baseline (Case 1). This suggests that while ToF's direct distance measurements may help correct for drift, the inherent noise in ToF sensors introduces high-frequency errors, or jitter, into the final pose estimation. The paper fails to discuss this drift-jitter trade-off, presenting a potentially incomplete and overly optimistic picture of the proposed method's performance.

**-** The contribution of Dyn-PE requires more explanation. The authors argue that static positional encodings, common in NLP, are ill-suited for wearable sensors whose spatial relationships are constantly changing. Their proposed solution is to make the positional value for each node a learned function of the entire sensor input at that timestep, i.e., PE_dyn (t) is a function of X_all (t). This approach raises a fundamental question: if the positional encoding already depends on the full input, what new relational information is the Transformer's self-attention mechanism expected to learn? The core purpose of self-attention is precisely to compute dynamic, input-dependent relationships between tokens. Dyn-PE appears to be pre-computing a simplified version of this relationship and adding it to the input, which feels redundant. While the ablation study in Table 3 shows a marginal improvement in some metrics (e.g., Positional Error on ToF-IP-DB drops from 7.36 to 6.31), this result is presented without any statistical validation. It is therefore impossible to know whether this small gain is a genuine contribution of the architecture or simply a product of random initialization or training dynamics. The claim that this technique "enhances spatial awareness" requires more rigorous proof to be considered a significant contribution.

---

> ### Author Rebuttal · Authors · 2025-07-31
>
> Thank you for your constructive feedback! We carefully address each point below, grouped by the original comments. For brevity, we refer to Weaknesses, Questions and Limitations as W#, Q# and L# respectively. All major clarifications and revisions will be incorporated into the final manuscript.
>
> W1, Q1: We trained our model 10 times with different random seeds and recorded the mean, std and the 95% confidence intervals for each metric.
> The results for the ablation study are as follows:
> | DIP     | SIP Global Error (°)        | Angular Global Error (°)    | Positional Error (cm)    |Endpoint Error (cm) | Jitter (1000 m/s³) |
> |----------|-----------------------------|-----------------------------|--------------------------|---------------------------|-------------------------------|
> | Case 1   | 18.94 ± 0.13 (18.84, 19.03) | 13.18 ± 0.17 (13.06, 13.30) | 6.86 ± 0.07 (6.80, 6.91) |  12.31 ± 0.11 (12.23, 12.39)   |0.07 ± 0.00 (0.07, 0.07) |
> | Case 2   | 18.98 ± 0.18 (18.86, 19.11) | 12.85 ± 0.11 (12.77, 12.93) | 6.94 ± 0.06 (6.90, 6.98) |  12.37 ± 0.10 (12.29, 12.44)   |0.13 ± 0.01 (0.13, 0.14)|
> | Case 3   | 17.91 ± 0.12 (17.83, 18.00) | 13.19 ± 0.08 (13.13, 13.25) | 6.81 ± 0.04 (6.78, 6.84) |   12.06 ± 0.08 (12.01, 12.12)   |0.13 ± 0.00 (0.13, 0.13)|
> | Case 4   | 17.21 ± 0.08 (17.16, 17.27) | 11.93 ± 0.06 (11.89, 11.97) | 6.31 ± 0.03 (6.29, 6.33) |  11.47 ± 0.06 (11.42, 11.52)   |0.12 ± 0.00 (0.12, 0.13) |
>
>
> | ToF-IP-DB  | SIP Global Error (°)         | Angular Global Error (°)      | Positional Error (cm)         |    Endpoint Error (cm)  |  Jitter (1000 m/s³)        |
> |--------|-------------------------------|-------------------------------|-------------------------------|-------------------------------|-------------------------------|
> | Case 1 | 16.34 ± 0.02 (16.33, 16.36)   | 7.68 ± 0.04 (7.65, 7.71)      | 5.79 ± 0.06 (5.75, 5.84)      |    8.44 ± 0.09 (8.37, 8.51)      | 0.13 ± 0.01 (0.12, 0.13)  |
> | Case 2 | 16.13 ± 0.08 (16.07, 16.19)   | 7.53 ± 0.07 (7.48, 7.57)      | 5.49 ± 0.04 (5.46, 5.51)      |    8.10 ± 0.06 (8.05, 8.14)      |0.18 ± 0.01 (0.17, 0.18) |
> | Case 3 | 15.49 ± 0.11 (15.41, 15.56)   | 7.29 ± 0.07 (7.25, 7.34)      | 5.30 ± 0.03 (5.28, 5.32)      |    7.79 ± 0.07 (7.74, 7.83)      |0.17 ± 0.01 (0.16, 0.17) |
> | Case 4 | 13.62 ± 0.42 (13.32, 13.92)   | 6.82 ± 0.16 (6.71, 6.93)      | 4.59 ± 0.09 (4.52, 4.65)      |    6.67 ± 0.13 (6.58, 6.76)      |0.17 ± 0.01 (0.16, 0.17)  |
>
>
> Due to the time constraints of the rebuttal period, we can only provide the results for the comparison with the closest SOTA:
> |  DIP  | SIP Global Error (°)         | Angular Global Error (°)      | Positional Error (cm)         |Endpoint Error (cm)           | Jitter (1000 m/s³)       |
> |-----------|-------------------------------|-------------------------------|-------------------------------|-------------------------------|-------------------------------|
> | DynaIP| 13.85 ± 0.46 (13.49, 14.21)   | 7.00 ± 0.18 (6.87, 7.13)   | 5.01 ± 0.11(4.93, 5.09)      | 6.82 ± 0.12(6.78, 6.90)      | 0.12 ± 0.00 (0.12, 0.13)      |
> | OURS  | 13.62 ± 0.42 (13.32, 13.92)   | 6.82 ± 0.16 (6.71, 6.93)      | 4.59 ± 0.09 (4.52, 4.65)      | 6.67 ± 0.13 (6.58, 6.76)     | 0.17 ± 0.01 (0.16, 0.17)      |
>
> | TOF-IP-DB   | SIP Global Error (°)         | Angular Global Error (°)      | Positional Error (cm)         | Endpoint Error (cm)           | Jitter (1000 m/s³)       |
> |-----------|-------------------------------|-------------------------------|-------------------------------|-------------------------------|-------------------------------|
> | DynaIP  | 19.00 ± 0.10 (18.93, 19.07)  | 13.38 ± 0.07 (13.33, 13.44)  |  7.22 ± 0.04 (7.19, 7.26)     | 13.02 ± 0.08 (12.97, 13.08)     |   0.16 ± 0.01 (0.15, 0.16)    |
> | OURS    | 17.21 ± 0.08 (17.16, 17.27)   | 11.93 ± 0.06 (11.89, 11.97)      | 6.31 ± 0.03 (6.29, 6.33)      |11.47 ± 0.06 (11.42, 11.52)      | 0.12 ± 0.00 (0.12, 0.13)     |
>
> We will provide a full statistic analysis of other SOTAs in revision.
>
> W2: To clarify, the slight increase in jitter introduced by ToF is not due to the inherent sensor noise, but rather the numerical uncertainty of distance map pixels when they correspond to the edge regions of the observed object. In fact, jitter caused by sensor data noise is a common challenge in wearable motion capture systems. For jitter reduction, a practical solution is to apply motion optimization to post-process the raw motion output. As shown below, after applying motion optimization in PNP, the jitter decreases significantly—from 0.12 m/s³ to 0.08 m/s³—ensuring the quality of motion capture in practical applications.
>
> W3: We would like to highlight the crucial role of Dyn-PE to the value matrix V in self-attention: it injects dynamic positional information into token values, which cannot be achieved with predefined or learnable static positional encodings.
> Also, as detailed in our response to W1, Q1, the statistical analyses of Case 2 and Case 4 (ablation study) across 10 random seeds show that Dyn-PE consistently improves performance.
>
> Q2: Due to space constraints, please refer to our response to Reviewer 2RCo, W2 for details. The results show that our model’s performance degrades gradually, without catastrophic failure, demonstrating its robustness to partial ToF sensor loss.
>
> Q3: This trade-off has already been addressed in detail in W2 of our response.
>
> Q4: As shown in the table below, we compared depth data collected from the real ToF hardware prototype and simulated ToF data generated under the same pose. Specifically, we selected representative frames from the left hand and left foot, and extracted the corresponding 4×4 pixel regions from both the real and simulated distance maps. The simulated ToF distance values were generated based on the same skeletal motion used during real data capture.
>
> Left Ankle
> | Simulated Distance（mm） | Real Distance（mm） |
> | ------------- | ------------- |
> | 2000          | 2000          |
> | 2000          | 2000          |
> | 2000          | 2000          |
> | 2000          | 2000          |
> | 2000          | 2000          |
> | 2000          | 2000          |
> | 2000          | 2000          |
> | 2000          | 2000          |
> | 376           | 369           |
> | 337           | 361           |
> | 337           | 369           |
> | 353           | 337           |
> | 235           | 235           |
> | 235           | 235           |
> | 227           | 227           |
> | 227           | 219           |
>
>
> Left Wrist
> | Simulated Distance（mm） | Real Distance（mm） |
> | ------------- | ------------- |
> | 227        | 125        |
> | 243        | 133        |
> | 243        | 133        |
> | 282        | 251        |
> | 243        | 133        |
> | 243        | 125        |
> | 243        | 125        |
> | 204        | 188        |
> | 2000       | 2000       |
> | 243        | 133        |
> | 227        | 133        |
> | 133        | 149        |
> | 1969       | 2000       |
> | 2000       | 2000       |
> | 243        | 133        |
> | 125        | 133        |
>
>
> Based on this quantitative comparison, the average absolute difference across the 16 pixels is 7.2 mm, this result indicates that the simulated ToF data closely approximates real sensor outputs under similar motion conditions.
>
> L1: We agree that optical ToF sensors may degrade under extreme conditions. However, in our tests, our system performed reliably on typical indoor and outdoor surfaces (e.g., tile floors, dark carpets, and asphalt), where ToF sensors returned stable and valid measurements at a true distance of 1000 mm across these scenarios.
>
> | Scene                          | Mean Distance(mm) | Std(mm) |
> |-------------------------------|--------------|-------------|
> | Office(Ceramic tile)        | 991.36       | 17.67       |
> | Office(Dark Carpet)         | 999.63       | 36.82       |
> | Outdoor(Sunny Grass)         | 1005.00      | 51.63       |
> | Outdoor(Shaded Grass)         | 999.34       | 23.59       |
> | Outdoor(Sunny Asphalt Road)     | 1004.37      | 80.91       |
> | Outdoor(Shaded Asphalt Road)      | 998.37       | 15.81       |
>
>
> In addition, we hope to clarify that the core contribution of our work is to demonstrate that integrating absolute distance measurements help IMU-based motion capture. ToF sensing is currently the most practical and accessible means to achieve this. As emerging MEMS-based or alternative distance sensors mature, they can be seamlessly integrated into our framework to address other challenging scenarios, with little or no changes to our algorithm.
>
> L2: We exported the model to ONNX and tested it on an Intel i7-13700K CPU. It achieved 0.65 ± 0.69 ms per frame (311.73 ± 78.34 FPS) over 18,000 frames. To ensure temporal consistency with time-differentiated acceleration, we cap the runtime at 60 FPS, under which latency remains 1.10 ± 1.05 ms and measured FPS is 60.44 ± 4.57.
> These results confirm real-time performance on a standard CPU, supporting deployment on portable or embedded systems without a GPU.
>
> L3: Our system samples IMUs at 100 Hz and ToF sensors at 60 Hz, using an MCU-based hardware timer to trigger data acquisition: 10 ms for IMUs and 16.66 ms for ToF. Hardware-level triggering minimizes jitter and ensures stable sampling intervals.
> For post-processing, we apply linear interpolation to align IMU, ToF, and motion capture ground truth to a unified 60 Hz timeline, ensuring consistent timestamps and low-latency alignment across modalities.

---

> > ### Comment · Reviewer_K3tx · 2025-08-04
> >
> > I am not satisfied with the rebuttal. I will keep my score.

---

### Note · Authors · 2025-08-13

We sincerely thank all reviewers for their thoughtful evaluations and constructive feedback throughout the review process. We greatly appreciate the recognition of our work’s contributions across hardware, algorithm design, and dataset release. In particular, we are encouraged by comments highlighting that:

**Innovative sensing integration in a practical design**: This work presents the first exploration of fusing sparse Time-of-Flight (ToF) distance maps with a standard 6-IMU motion capture setup to directly mitigate cumulative error inherent to IMU-only systems, realized in a compact, wearable prototype integrating four ToF sensors with minimal size and weight increase, preserving comfort and mobility. (K3tx)

**Clear Problem Definition and Strong Motivation**: The work addresses a fundamental and persistent problem in wearable motion capture — the quadratic error accumulation from double integration of noisy accelerometer data in IMU-only systems. The limitations of purely IMU-based approaches (integration andcumulative errors) are clearly explained, which provides a compelling motivation for the proposed solution. (K3tx, 2RCo)

**Valuable dataset contribution**: The release of ToF-IP-DB, a 208-minute synchronized ToF–IMU–optical dataset, was acknowledged by reviewers K3tx and 2RCo as a significant asset for the research community.

**Advancing Structured Sensor Fusion and Dynamic Spatial Modeling**: The proposed Node-Centric Data Integration (NCI) and Dynamic Spatial Positional Encoding (Dyn-PE) were highlighted as principled methods to address the structured multi-sensor integration and dynamic spatial awareness challenges in wearable motion capture. (2RCo, zeff)

**Comprehensive experiments**: Ablation studies, robustness tests under partial occlusion, and SOTA comparisons were noted as thorough and relevant. Our method delivers “state-of-the-art results” with “novel sensors for pose estimation”. (2RCo, 8u8c)

We are pleased that reviewers recognized the technical soundness, practical relevance, and potential impact of our contributions, and we will incorporate all agreed-upon improvements into the revision.

---

### Decision · Program_Chairs · 2025-09-17

**Decision:**

Accept (poster)

**Comment:**

This paper received 2 Borderline Accept, 1 Accept and 1 Borderline Reject.
The paper shows how the integration of ToF sensors and IMU improves pose tracking, in particular it addresses the typical drift of double integration in IMU trackers.
The main concern of the negative reviewer was a lack of ethics form, but authors confirmed subjects signed and ethics form and are compliant with ethics guidelines in their institution but did not upload the ethics form due to anonymity.
Reviewers overall acknowledge the novelty, the dataset introduced and the quality of results overall. The remaining concerns such as lack of results and ablations are not grounds for rejection.
The AC recommends acceptance.